# Rapid microbial methanogenesis during $CO_2$ storage in hydrocarbon reservoirs

R. L. Tyne[1✉], P. H. Barry[1,2✉], M. Lawson[3,9✉], D. J. Byrne[4], O. Warr[5], H. Xie[6], D. J. Hillegonds[1], M. Formolo[7], Z. M. Summers[8], B. Skinner[7], J. M. Eiler[6] & C. J. Ballentine[1✉]

Carbon capture and storage (CCS) is a key technology to mitigate the environmental impact of carbon dioxide ($CO_2$) emissions. An understanding of the potential trapping and storage mechanisms is required to provide confidence in safe and secure $CO_2$ geological sequestration[1,2]. Depleted hydrocarbon reservoirs have substantial $CO_2$ storage potential[1,3], and numerous hydrocarbon reservoirs have undergone $CO_2$ injection as a means of enhanced oil recovery ($CO_2$-EOR), providing an opportunity to evaluate the (bio)geochemical behaviour of injected carbon. Here we present noble gas, stable isotope, clumped isotope and gene-sequencing analyses from a $CO_2$-EOR project in the Olla Field (Louisiana, USA). We show that microbial methanogenesis converted as much as 13–19% of the injected $CO_2$ to methane ($CH_4$) and up to an additional 74% of $CO_2$ was dissolved in the groundwater. We calculate an in situ microbial methanogenesis rate from within a natural system of 73–109 millimoles of $CH_4$ per cubic metre (standard temperature and pressure) per year for the Olla Field. Similar geochemical trends in both injected and natural $CO_2$ fields suggest that microbial methanogenesis may be an important subsurface sink of $CO_2$ globally. For $CO_2$ sequestration sites within the environmental window for microbial methanogenesis, conversion to $CH_4$ should be considered in site selection.

A possible method for reducing current greenhouse gas emission rates is carbon capture and storage (CCS). Long-term carbon dioxide ($CO_2$) trapping mechanisms include structural or stratigraphic trapping in stable geological configurations, dissolution into pore fluids (solubility trapping)[4], carbonate mineralization (precipitation and mineral trapping)[5] or adsorption (for example, onto coal)[6]. Typically, during $CO_2$ enhanced oil recovery ($CO_2$-EOR), a proportion of the injected $CO_2$ remains within the reservoir post-injection[3], providing an analogue for investigating and quantifying processes within CCS sites, over decadal timescales.

Here we investigate the behaviour of $CO_2$ within the Olla Oil Field, Louisiana, USA (Fig. 1), which was $CO_2$-EOR flooded in the 1980s, and compare its (bio)geochemical composition with the adjacent Nebo-Hemphill Oil Field, which was never subjected to $CO_2$-EOR (Methods). Approximately $9 \times 10^7$ $m^3$ (standard temperature and pressure (STP)) of injected $CO_2$ has been retained in the Olla reservoirs post-injection[7]. The injected $CO_2$ was sourced from the Black Lake Oil Field, which is located adjacent to the Sabine Island Complex, a basement high and likely conduit for mantle-derived fluids[8]. Although the Olla Oil Field has been geochemically characterized by several previous studies[9–11] (Methods), this study integrates noble gas and clumped isotope data with stable isotope and microbiological data to investigate the fate of the injected $CO_2$. Previous studies have suggested that microbial hydrocarbon degradation and methanogenesis may occur

within both the Olla and Nebo-Hemphill oil fields[9–11]. Microbial methanogenesis takes acetate, methylated compounds, or hydrogen and $CO_2$ (hydrogenotrophic methanogenesis, equation (1)) as its starting point; the latter mechanism is most relevant to this study. Hydrogenotrophic methanogenesis results in [13]C enrichment of residual $CO_2$ and the formation of methane ($CH_4$) initially [13]C depleted compared with the source $CO_2$, which increases to the value of the consumed $CO_2$.

$$CO_2 + 4H_2 \rightarrow CH_4 + 2H_2O \tag{1}$$

Given their proximity and comparable geological histories (including hydrocarbon production from similar reservoirs; Methods), we assume that the Nebo-Hemphill and Olla oil fields initially had comparable geochemical compositions, before $CO_2$-EOR at Olla. For example, the Olla and Nebo-Hemphill oil fields have broadly overlapping pH, temperatures, salinities and ion chemistry (Extended Data Table 1). Despite $CO_2$ injection ceasing in 1986, $CO_2$ concentrations within the Olla Oil Field ($18 \pm 12$ mol% (STP) across all reservoirs) are greater than breakthrough concentrations limits (10%)[7] and significantly greater than those measured at Nebo-Hemphill ($1.02 \pm 0.69$ mol% (STP)). We observe a higher $\delta^{13}C_{VPDB}$ of $CO_2$ at Olla ($13.5 \pm 3.4$‰) compared with both Nebo-Hemphill ($4.8 \pm 4.9$‰) and the injected $CO_2$ ($0.85 \pm 0.92$‰)[10] (Extended Data Table 2), where $\delta^{13}C = [(^{13}C/^{12}C)_{sample}/(^{13}C/^{12}C)_{standard}] - 1$ and VPDB is Vienna PeeDee Belemnite. Injection alone cannot account

[1]Department of Earth Sciences, University of Oxford, Oxford, UK. [2]Department of Marine Chemistry and Geochemistry, Woods Hole Oceanographic Institution, Woods Hole, MA, USA. [3]ExxonMobil Upstream Business Development, Spring, TX, USA. [4]CRPG-CNRS, Université de Lorraine, Nancy, France. [5]Department of Earth Sciences, University of Toronto, Toronto, Ontario, Canada. [6]Division of Geological and Planetary Sciences, California Institute of Technology, Pasadena, CA, USA. [7]ExxonMobil Upstream Integrated Solutions, Spring, TX, USA. [8]ExxonMobil Research and Engineering Co., Virginia, NJ, USA. [9]Present address: Aker BP, Stavanger, Norway. ✉e-mail: rebecca.tyne@earth.ox.ac.uk; pbarry@whoi.edu; miclawson@gmail.com; chris.ballentine@earth.ox.ac.uk

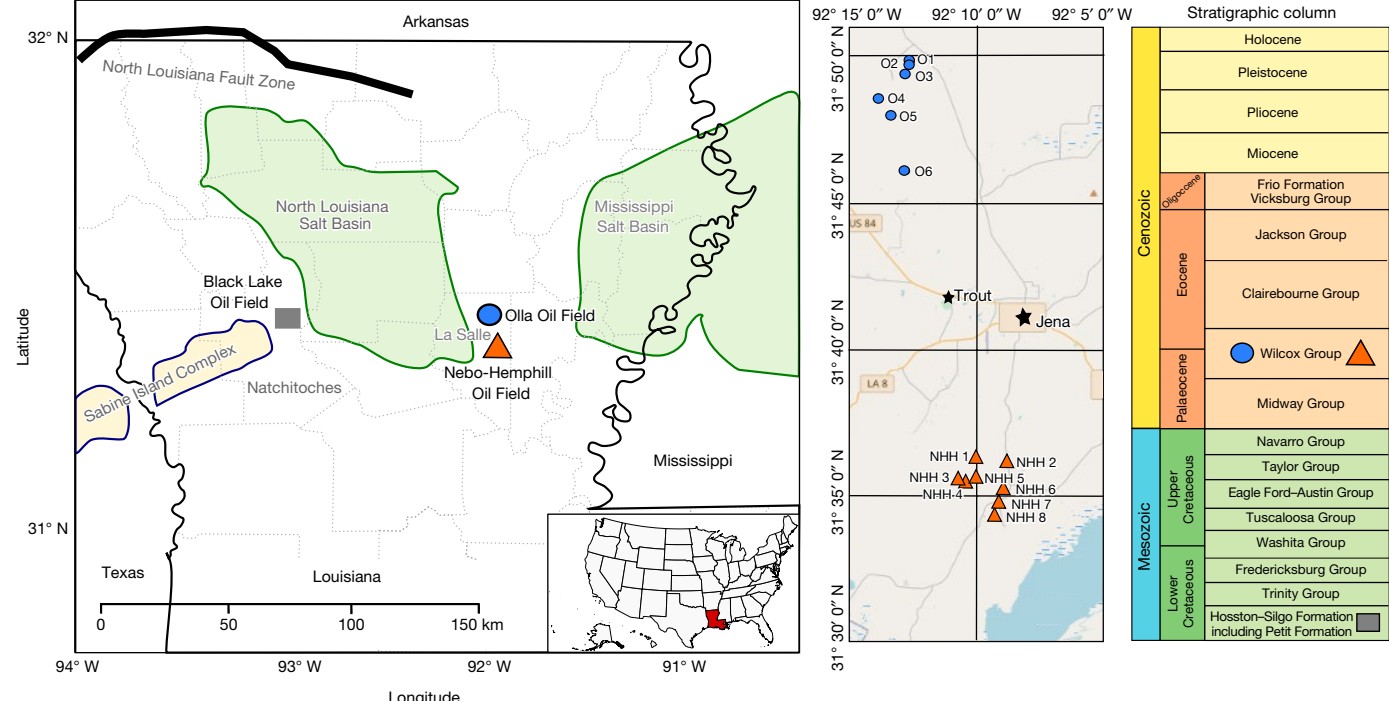

**Fig. 1 | Map of study area showing locations of the Olla and Nebo-Hemphill oil fields as well as the Black Lake Oil Field, from which the injected CO₂ was sourced.** Only the Olla Oil Field contains injected CO₂. The inset on the right shows an expanded view with individual sample locations (nearby urban areas are denoted by stars) and a stratigraphic column showing the relevant lithologic units[9].

for the higher $\delta^{13}C$ of $CO_2$ values at Olla; therefore, the integration of complementary datasets is required to evaluate additional in situ processes.

Owing to their inert nature, noble gases provide a powerful tool for tracing and quantifying physiochemical processes associated with $CO_2$ injection[4,8,12–19]. The average air-corrected helium isotope ($^3He/^4He$) ratio relative to the atmospheric ratio ($R/R_A$, where $R = {}^3He/{}^4He_{sample}$ and $R_A = {}^3He/{}^4He_{air}$) was determined for the Olla and Nebo-Hemphill oil fields to be $1.76 \pm 0.31\,R_A$ and $0.46 \pm 0.12\,R_A$, respectively (Extended Data Fig. 1, Extended Data Table 2). The elevated He isotope values in the Olla Oil

Field require an enhanced mantle-derived noble gases contribution, which is supported by additional noble gas data (Extended Data Fig. 1, Methods), associated with injected $CO_2$, which were sourced from the Black Lake Oil Field.

In addition to characterizing fluid origin, a combined noble gas and $CO_2$ isotope approach provides insight into processes associated with $CO_2$ trapping (for example, ref. [4]) or methanogenesis following injection. $^3He$ is inert and insoluble, with no significant sources within the crust. Thus, variations in $CO_2/^3He$ post-injection are directly attributable to the addition or removal of $CO_2$ within the system[4,20]. Although

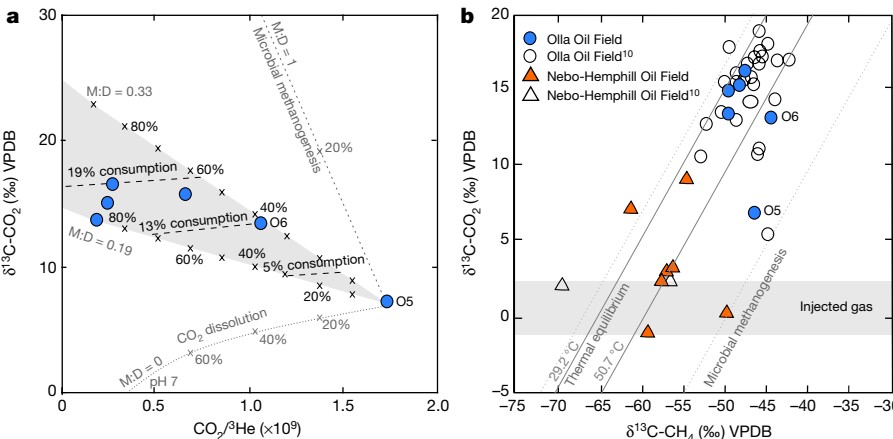

**Fig. 2 | The $\delta^{13}C$ of $CO_2$ in the Olla ($CO_2$ injected field) samples. a**, The $\delta^{13}C$ of $CO_2$ as a function of the $CO_2/^3He$ ratio. The dashed lines show endmember methanogenesis and dissolution (pH 7) fractionation trajectories. The tick marks represent the total amount of $CO_2$ trapping within the system, relative to sample O5. The shaded region represents trapping by the combination of both microbial methanogenesis and dissolution. The upper and lower methanogenesis:dissolution ratios (M:D) are 0.33 and 0.19, respectively, showing that dissolution accounts for approximately three times more $CO_2$

removal (M:D = 0.26) than microbial methanogenesis. The lines labelled 'consumption' show the portion of original injected $CO_2$ that has been removed by net microbial methanogenesis. **b**, The $\delta^{13}C$ of $CO_2$ as a function of the $\delta^{13}C$ of $CH_4$. The shaded region represents the $CO_2$ isotopic composition of the injectate from the Black Lake Oil Field into the Olla system. The Olla data are consistent with thermal re-equilibration with both reservoir temperatures (solid lines) and microbial methanogenesis (dotted lines). The $1\sigma$ level of uncertainty is within the symbol size.

two samples at Olla (samples O5 and O6) have $CO_2/^3He$ within the mantle range $(2 \pm 1 \times 10^9)$, consistent with the injection of mantle-derived fluids, the majority of the ratios are lower and indicate $CO_2$ removal (Fig. 2a, Extended Data Table 2). The difference between the highest $CO_2/^3He$ (O5, considered the most pristine sample) and the remaining samples provides a conservative estimate of post-injection $CO_2$ trapping/consumption of between 39% and 89%.

A decrease in the $CO_2/^3He$ alongside increasing $^4He$ and $^{20}Ne$ concentrations is observed within the Olla Oil Field (Extended Data Fig. 2, Methods). In the subsurface, $^4He$ is produced via radiogenic decay of uranium and thorium and accumulates in formation water[12], which also contains atmosphere-derived $^{20}Ne$. An anticorrelation between $CO_2/^3He$ and $^4He$ and $^{20}Ne$ is observed in natural CCS analogues globally[4], and indicates that the extent of water contact controls the magnitude of $CO_2$ trapping via dissolution or precipitation of $CO_2$ (refs. [4,15]). Within the observed pH range $(7.0 \pm 0.5)$ at Olla, dissolution will dominate over precipitation as a $CO_2$ trapping mechanism[4,10]. However, as described above, the $\delta^{13}C$ of $CO_2$ in the Olla gases is more enriched than can be explained by these processes alone (Fig. 2a).

Independent of the source of the injected $CO_2$, such positive $\delta^{13}C$ values suggest significant modification from a starting composition that is unlikely to be heavier than 2‰ (refs. [21,22]). Notably, sorption to coal, hydrocarbon biodegradation to $CO_2$, and subsequent consumption and mixing processes could result in an increase in the $\delta^{13}C$ of $CO_2$ (Methods). However, even from the most pristine sample (O5), it is unlikely that any of these processes could generate a fluid with the elevated concentrations of the isotopically enriched $CO_2$ observed at Olla.

Further constraints on the origins and processes affecting these gases are provided by the molecular-average and clumped isotope compositions of the associated $CH_4$. We observe a relatively high $\delta^{13}C$ of $CO_2$ $(13.6 \pm 3.2‰)$ and $CH_4$ $(-47.7 \pm 1.7‰)$ at Olla compared with those observed at Nebo-Hemphill $(3.5 \pm 3.6‰$ and $-56.4 \pm 3.6‰$ for $CO_2$ and $CH_4$, respectively) (Fig. 2b, Extended Data Fig. 3a, Extended Data Table 2). The higher $\delta^{13}C$ of $CH_4$ at Olla could be interpreted as evidence that the Olla Oil Field contains a higher proportion of thermogenic gas, or thermogenic gas of higher thermal maturity, whereas the Nebo-Hemphill Oil Field contains more microbial gas or thermogenic gas of lower thermal maturity. Using our multi-isotope approach, we present a model that is consistent with the observed higher $\delta^{13}C$ values of $CH_4$ and an increase in the $\delta^{13}C$ of residual $CO_2$. In this model, Olla is assumed to have had a pre-$CO_2$-EOR composition resembling Nebo-Hemphill, but following $CO_2$-EOR, microbial activity converted significant amounts of injected $CO_2$ to $CH_4$ (Extended Data Fig. 4). We arrive at this conclusion on the basis of the following observations.

First, we report a difference between the $\delta^{13}C$ of $CH_4$ and the $\delta^{13}C$ of $CO_2$ of 53.7–64.4‰ at the Olla Oil Field and of 50.8–68.1‰ at the Nebo-Hemphill Oil Field (Fig. 2b). The observed carbon isotope fractionations between co-existing $CO_2$ and $CH_4$ are consistent with thermodynamic isotopic equilibrium at temperatures between 44.6 °C and 76.3 °C at Olla and between 29.8 °C and 85.3 °C at Nebo-Hemphill[23], which overlap the present-day reservoir temperatures within these fields (29.2–50.7 °C and 29.7–57.1 °C, respectively), which suggests that the systems are approaching isotopic equilibrium under current reservoir conditions (Methods). The isotopic approach to equilibrium under reservoir conditions appears to be a result of microbial cycling of carbon (that is, methanogenesis and anaerobic oxidation of methane (AOM); evidence for AOM is apparent in the clumped isotopologues, see below). Equilibrium with reservoir conditions between the $\delta^{13}C$ of $CO_2$ and the $\delta^{13}C$ of $CH_4$ has previously been observed under similar geological conditions[24].

Second, we observe a range in the two measurable clumped isotopologues of $CH_4$, $\Delta^{13}CH_3D$ and $\Delta^{12}CH_2D_2$, at Olla of 3.45–5.62‰ and 9.13–12.4‰ (Fig. 3, Extended Data Table 2), respectively, which correspond to apparent temperature ranges of $29^{+14}_{-12}$ °C to $128^{+17}_{-16}$ °C and $87^{+13}_{-11}$ °C to

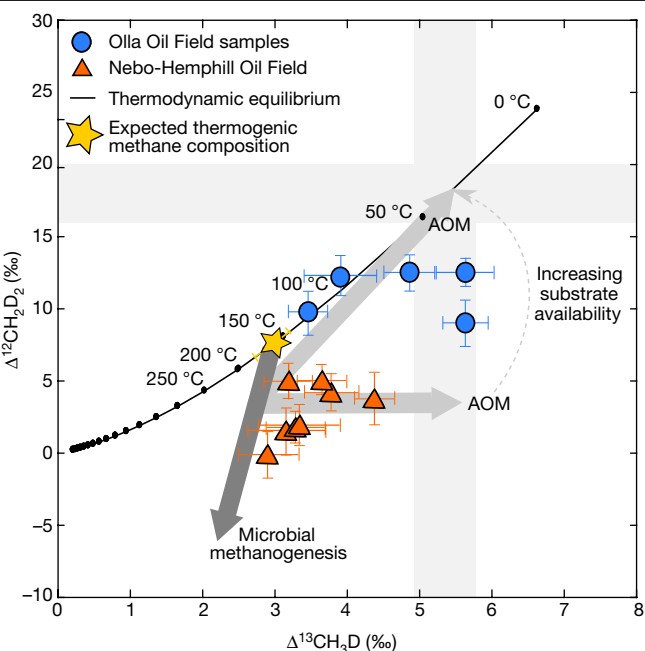

**Fig. 3 | $\Delta^{12}CH_2D_2$ versus $\Delta^{13}CH_3D$ of the measured Olla and Nebo-Hemphill samples.** The clumped isotopologue space illustrates whether measured $CH_4$ is at internal thermodynamic equilibrium (black line) or not. The thermodynamic equilibrium curve is calculated following ref. [41]. The shaded cross represents thermal equilibration to current reservoir temperatures in each isotopologue. The arrows represent the theoretical trends for methanogenesis (dark grey) and AOM (light grey)[29]. The $1\sigma$ level of uncertainty is shown on the measured samples.

$132^{+30}_{-23}$ °C, respectively, and are approaching equilibrium with current reservoir conditions. Clumped isotope compositions provide an independent constraint on the origin of $CH_4$ in petroleum systems[25–30]. Thermogenic $CH_4$ appears to dominantly form under internal isotopic equilibrium[26,30] whereas microbial $CH_4$ is highly variable in its clumped isotope compositions, with both non-equilibrium[28,29] and apparent equilibrium signatures[24,26,28,29,31–34]. For Olla, we expect thermogenic-$CH_4$-generation temperatures to have exceeded 163 ± 18 °C, from independent maturity constraints provided by biomarkers[35]. Two samples from Olla (O4 and O6) appear to be in internal isotopic equilibrium (about 125 °C and about 95 °C) but have lower apparent temperatures than expected for pure thermogenic $CH_4$. These temperatures are consistent with a component of microbial $CH_4$ formed at or close to equilibrium with current reservoir conditions. The remaining samples have $\Delta^{13}CH_3D$ apparent temperatures of $28.7^{+13.6}_{-12.3}$ °C to $74.0^{+16.8}_{-15.0}$ °C, within error of the present-day reservoir temperature for these fluids (Methods), consistent with in situ microbial methanogenesis and AOM. Furthermore, these samples exhibit deficits in $\Delta^{12}CH_2D_2$, consistent with fluids that are dominated by microbial $CH_4$[28,29,34] in a system approaching equilibrium via active methanogenesis and AOM. In contrast, the $\Delta^{13}CH_3D$-based temperatures of $77.9^{+13.5}_{-12.3}$ °C to $166^{+34}_{-29}$ °C at Nebo-Hemphill suggest that these fluids are dominated by thermogenic $CH_4$, although, similarly to Olla, the $\Delta^{12}CH_2D_2$ clumped isotopologues also suggest a microbial $CH_4$ contribution.

Third, we identify evidence for the biodegradation of hydrocarbons in the molecular geochemistry of oils produced from both the Olla and the Nebo-Hemphill oil fields (Methods). In the case of Olla, we also report an elevated $\delta^{13}C$ of propane (Extended Data Fig. 3b). The presence of oils co-existing with propane in these reservoirs preclude the possibility of high-temperature cracking as the cause of the positive $\delta^{13}C$ values and suggests that the propane has been subjected to biodegradation[36], consistent with low-temperature microbial activity within the field.

Fourth, small subunit (SSU) ribosomal RNA gene sequencing of microbial communities identifies the presence of hydrogenotrophic methanogens, methanol and methylotrophic methanogens, and anaerobic methanotrophs (ANME) assigned sequences (Extended Data Fig. 5). These were sampled from the formation water in two wells within each field alongside previous microbiology analysis of these fields[37]. Co-occurrence of both methanogenic and methanotrophic Archaea shows the potential for conversion of $CO_2$ to $CH_4$ and vice versa, consistent with isotopic signatures at intramolecular and intermolecular equilibrium. However, equilibration is probably rate limited by AOM, as the abundance of methanogens was 100 times higher than ANME, and AOM probably proceeds at a lower rate per cell.

Combined independent geochemical and microbiological data support our assertion that oil field microbes convert injected $CO_2$ to $CH_4$ at Olla. This study identifies the microbial processes associated with the injection of $CO_2$ into the deep subsurface operating on decadal timescales and the scale of the processing.

To quantify the impact of these microbial processes on the Olla Oil Field, we have constructed a model that considers both dissolution and microbial methanogenesis (Fig. 2a, Methods). From our most pristine sample (O5, highest $CO_2/^3He$), we find that to match the isotopic composition of $CO_2$ with its corresponding $CO_2/^3He$, dissolution must account for $3.1 \pm 1.1$ times more $CO_2$ removal than hydrogenotrophic methanogenesis. From this model, net microbial methanogenesis is estimated to have consumed a minimum of between 13% and 19% of the post-injection $CO_2$, depending on sample location, whereas dissolution is responsible for removal of as much as 74% more $CO_2$. This estimate is independently confirmed when $CO_2$ concentrations are compared with the $\delta^{13}C$ of $CO_2$ values (Extended Data Fig. 6). On the basis of the elevated $\delta^{13}C$ of $CO_2$, we recognize that our most pristine sample has probably also undergone a significant degree of modification, consistent with our estimate being conservative, and that $CO_2$ concentrations post-injection were likely to be higher and variable across the field.

Our combined noble gas and stable isotope approach allows estimates of in situ net microbial methanogenesis rates from within a natural system to be calculated. By extrapolating our results over the 29 years between the cessation of injection (1986) and sampling (2015), and assuming 13–19% microbial consumption of $CO_2$ since injection (from the hybrid methanogenesis–dissolution model, Fig. 2) with the remaining injected $CO_2$ volume, we calculate that a minimum of $1.15 \times 10^7$–$1.72 \times 10^7\ m^3$ (STP) of microbial $CH_4$ has been produced at a minimum rate of 73–109 mmol $CH_4\ m^{-3}$ (STP) $yr^{-1}$. Previous estimates for $CO_2$ reduction following methanogenic oil degradation by hydrogenotrophic methanogens in lab microcosm incubations at similar temperatures are significantly lower (about 0.01–0.15 mmol $CH_4\ m^{-3}$ (STP) $yr^{-1}$)[38] by comparison.

This identification and quantification of $CO_2$-EOR-related microbial methanogenesis at Olla has significant implications, and warrants reconsideration of reservoir processes at similar sites worldwide. The fate of injected $CO_2$ in other oil fields (for example, Belridge Diatomite Formation, Lost Hills Oil Field, California, USA[17,39]) has never been systematically explored using this integrated approach. Similarly to Olla, the Lost Hills Field has an elevated $\delta^{13}C$ of dissolved inorganic carbon (DIC) (20.16–23.61‰)[40] compared with the injected $\delta^{13}C$ of $CO_2$ (−30.1‰)[39]. The fractionation between $\delta^{13}C$ of DIC and $CH_4$ suggests that the Lost Hills system is approaching equilibrium with current reservoir conditions and the inverse correlation between the $\delta^{13}C$ of the post-injection residual $CO_2$ and concentration is consistent with microbial methanogenesis. Furthermore, data from Nebo-Hemphill, as well as naturally $CO_2$-rich hydrocarbon systems in the Pannonian Basin, Hungary, are also consistent with significant microbial methanogenesis based on the $CO_2/^3He$ and the $\delta^{13}C$ of $CO_2$ (Extended Data Fig. 7, Methods). These examples suggest that hydrogenotrophic methanogenesis probably occurs in both natural and injection-perturbed systems when reservoirs have favourable physiochemical conditions.

We conclude that although methanogenesis is not the dominant $CO_2$ sink, it can represent a substantial process within both natural and perturbed $CO_2$ fields and may be significantly more prevalent than previously considered. Even with the most conservative estimates of $CO_2$ conversion by methanogenesis in the Olla Oil Field, we find that, so far, as much as 13% to 19% of the emplaced $CO_2$ in sampled sections of the field has been consumed by methanogens. This is less than the amount of $CO_2$ that has been dissolved into water, but similarly occurs at significant rates on engineering timescales. $CH_4$ is less soluble and more mobile than $CO_2$ and therefore there is an enhanced risk of gas loss associated with $CH_4$ production due to microbial methanogenesis. Depleted hydrocarbon reservoirs have the second-largest $CO_2$ storage potential[1,3] after deep saline aquifers. If physiochemical and environmental conditions within these reservoirs (for example, low temperature, low salinity and high substrate availability) are conducive to microbial methanogenesis, we suggest that these microbial processes should be considered as criteria for future CCS site selection to ensure low-risk long-term storage. In addition, integrated studies like this may prove essential to effectively monitor $CO_2$ storage and the biogeochemical processes that result as a consequence of it.

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

# Methods

## Modelling

Here we provide extra details of the $\delta^{13}C$-$CO_2$ fractionation modelling of methanogenesis and dissolution. In our models (Fig. 2a, Extended Data Fig. 6), we use the sample with the highest measured $CO_2$/$^3$He (most pristine sample, O5) as a reference point to calculate the change in $CO_2$/$^3$He and $\delta^{13}C$-$CO_2$ with $CO_2$ processing (that is, methanogenesis and dissolution). We assume there is no $^3$He loss from the gas phase by dissolution or methanogenesis and therefore changes in the fraction of the $CO_2$ cause changes in the $CO_2$/$^3$He.

Dissolution is modelled as open-system Rayleigh fractionation[4].

$$\delta^{13}C_{CO2} = \delta^{13}C_{CO2i} \times f^{(\alpha-1)}, \qquad (2)$$

where $f$ is the fraction of the original $CO_2$ remaining in the reservoir, $\delta^{13}C_{CO2}$ and $\delta^{13}C_{CO2i}$ are the carbon isotopic composition of the $CO_2$ for post-dissolution and in the initial $CO_2$ respectively, and $\alpha$ is the fractionation factor. The fractionation factor for dissolution ($\alpha_d = 1.0038 \pm 0.0012$) was calculated based on the average measured pH[10]. Precipitation curves (Fig. 2a) can be modelled in the same way using fractionation factors ($\alpha_p = 1.0086 \pm 0.0012$) calculated for the current reservoir temperature[21].

From consideration of the carbon isotopes, which appear to be approaching equilibrium under current reservoir conditions, net methanogenesis has been modelled as a closed-system equilibrium process (equation (2), Fig. 2a). The fractionation resulting from this cycling, $\alpha_m$, is $0.9363 \pm 0.003$, which is consistent with thermal equilibration at current reservoir temperatures[23].

$$\delta^{13}C_{CO2} = \delta^{13}C_{CO2i} - (\alpha_m - 1)(1 - f) \times 1{,}000 \qquad (3)$$

Within the Olla Oil Field, it is likely that there is a combination of both microbial cycling and dissolution. As a result, the net effect of microbial methanogenesis and dissolution can be described using equation (3).

$$\delta^{13}C_{CO2} = \delta^{13}C_{CO2i} \times f^{(\alpha-1)} - (\alpha_m - 1)(1 - f) \times 1{,}000 \qquad (4)$$

where $f$ is the fraction of $CO_2$ remaining after trapping/consumption and $\alpha_A$ is the apparent fractionation factor for the system. $\alpha_A$ depends on the degree of dissolution compared with methanogenesis and is described as[42]:

$$\alpha_A = F_d \alpha_d + (1 - F_d)\alpha_m \qquad (5)$$

where $F_d$ is the fraction of $CO_2$ of the total $CO_2$ trapped/consumed due to dissolution.

## Sample collection and analysis

Samples for bulk gas composition, stable isotope, noble gas and clumped isotope analysis were collected directly from the wellhead using standard techniques following methods from refs. [8,17,18,24]. Noble gas isotope determination was conducted in the Noble Laboratory at the University of Oxford, where the analysis of hydrocarbon gases is well established[8,17,18,43–45]. For bulk composition and C and H stable isotope analysis, gas cylinders were shipped to Isotech in Champaign, Illinois, USA. The standard procedures for these are described in ref. [46]. $CH_4$ clumped isotopologues were determined at the California Institute of Technology following the procedures described in ref. [46].

Microbiological analysis was conducted at an internal ExxonMobil Research and Engineering Company Facility. The produced water samples were collected and filtered immediately from wells O1, O4, NHH3 and NHH6. Water was passed through 0.22-µm SVGPL10 RC Sterivex GP filters (EMD Milipore) until filters clogged in the following amounts: O1, 1,350 ml; O4, 1,350 ml; NHH3, 1,750 ml; NHH4, 300 ml. Filters were stored on ice immediately and stored at −80 °C until DNA extraction. DNA was extracted from the Sterivex filters using the DNeasy PowerWater Sterivex Kit (Qiagen). The SSU rRNA gene was amplified with V4V5 specific primers (https://vamps2.mbl.edu/) targeting Bacteria. The SSU rRNA gene was amplified with specific primers targeting Bacteria: forward primers (967F) CTAACCGANGAACCTYACC, CNACGCGAAG AACCTTANC, CAACGCGMARAACCTTACC and ATACGCGARGAACCTT ACC; reverse primer (1064R) CGACRRCCATGCANCACCT. A second set of amplifications was performed using specific primers targeting Archaea: (Arch2A519F) CAGCMGCCGCGGTAA and (Arch1017R) GGCC ATGCACCWCCTCTC. Illumina MiSeq 2x300bp paired-end sequencing was performed by Mr. DNA. Microbiome bioinformatics were performed with QIIME 2 2017.4[47]. Raw sequence data were demultiplexed and quality filtered using the q2-demux plugin followed by denoising with DADA2[48]. Amplicon sequence variants were aligned with mafft and used to construct a phylogeny with fasttree2[49,50]. Taxonomy was assigned to amplicon sequence variants using the q2-feature-classifier[51] classify-sklearn Naïve Bayes taxonomy classifier against the Greengenes 13_8 99% operational taxonomic unit (OTU) reference sequences[52].

## Geological and production history

The Olla Oil Field is located in La Salle Parish, Louisiana, in the northern Gulf of Mexico and was $CO_2$ injected between 1983 and 1986 for enhanced oil recovery ($CO_2$-EOR). The Nebo-Hemphill Oil Field lies 20 km to the southeast of the Olla Oil Field and provides a control field for this study, having had no $CO_2$ injected (Fig. 1). Both fields are producing from the Palaeocene–Eocene Wilcox Group, which in northern Louisiana is composed of 600–1,500 m of shallow marine clastic sediments to fluvial deltaic deposits and up to 6-m-thick coal beds[53,54]. The group is bound either side by the Palaeocene Midway Group and the Eocene Clairborne Group confining units[54]. Typically, the Wilcox Group is subdivided into the lower (sandstone with coal beds), middle (sandstone, lignite and clay) and upper (deltaic deposits) intervals, with the 'Big Shale' informal unit separating the middle and upper groups[53,55]. Late Cretaceous intrusive igneous bodies in northern Louisiana, west-central Mississippi and southeastern Arkansas are thought to have affected the regional heat flow, which may have caused an increase in the thermal maturity of the Wilcox Group during the early Palaeocene[56,57].

Gas/oil ratios are variable within the field, and do not seem to be associated with proximity to the known gas cap and higher gas/oil ratios may instead reflect gas coming out of solution as bottom hole pressure decreases.

$CO_2$ is present in low concentrations ($1.02 \pm 0.69$ mol%) within the Nebo-Hemphill Oil Field, which suggests that there is some natural background $CO_2$ present in both fields. Between March 1983 and April 1986, the 2,800-ft sandstone (within the middle Wilcox Group) in the Olla Oil Field underwent a pilot project for $CO_2$ flooding. The $CO_2$ injected into the field was immiscible and injected in the gas phase (that is, not supercritical). It is assumed that a negligible amount of $CH_4$ was injected alongside the $CO_2$. Previous studies on the Olla Oil Field have assumed that all the injected $CO_2$ was retained within the 2,800-ft sandstone[9,10]. However, the initial $CO_2$ injection report[7] states that "mole percentages in excess of 10% more than likely indicate production of injected $CO_2$". Only 1 sample outside of the 2,800-ft sandstone has $CO_2$ concentrations less than 10% (that is, all other $CO_2$ concentrations are greater than 10%). Notably, concentrations greater than 10% in other producing zones have previously been reported[10]. These signatures are consistent with injected $CO_2$ migration across the different reservoirs and remaining within the formations upon sampling. If such high concentrations of $CO_2$ were present in this field before injection, $CO_2$ would not have been extracted and transported from the Black Lake Oil Field and would instead have been extracted from the producing fluids of $CO_2$ rich fluids at Olla. In addition, we see a clear mantle-derived noble gas signature (see below),

which is most evident in the He and neon (Ne) isotopes (Extended Data Fig. 1), in the Olla Oil Field, which shows a clear incursion of injection $CO_2$ into the proximal lithologies.

A total of $2.2 \times 10^8$ $m^3$ of $CO_2$ was injected into the Olla Oil Field and was sourced from the nearby Black Lake Oil Field (produced from the Lower Cretaceous carbonates of the Petit Formation) in Natchitoches Parish, Louisiana (Fig. 1). Approximately one-third of the injected $CO_2$ remained within the reservoir after the cessation of the EOR project[7]. It is important to note that given the age of this $CO_2$-EOR project, we were unable to obtain constraints such as the ratio of injected $CO_2$ to reservoir pore volume that are sometimes cited to quantify the scale of the EOR project and the distribution it achieves within the subsurface. Further, we anticipate that the injected $CO_2$ is not uniformly distributed within the pore volume of the Olla Oil Field due to different phases of EOR and gas cycling.

In 1984, water flooding also started within the field. In contrast, Nebo-Hemphill has never undergone any type of EOR.

## Molecular geochemistry
Whole-oil gas chromatography (WOGC) was performed to provide a hydrocarbon fingerprint of each oil, and open-column liquid chromatography was utilized to separate and quantify the saturate, aromatic, resins and asphaltene fractions of each oil. In general, we see similar fluid compositions between the two fields. The WOGC shows evidence of marine-sourced oils with a smooth distribution of $n$-alkanes, pristine/phytane ratios of approximately 2 and low wax content. The notable differences are probably due to varying levels of biodegradation illustrated by the loss of $n$-alkanes in some samples and American Petroleum Institute (API) gravity ranging from 21 to 42. The similarities in the geochemical signatures of these oils provide a basis to use the Nebo-Hemphill Oil Field as a control site for any biogeochemical differences that may arise as a result of the injection of $CO_2$ at Olla.

## Source facies
Gas chromatography mass spectroscopy (GC/MS) was utilized to analyse the saturate and aromatic fractions to provide the biomarker distributions of each oil. The interpretation of a dominantly marine source is based on the following evidence: (1) a smooth distribution of tricyclic terpanes with $C_{23}$ being the maximum along with extended hopanes ($C_{31}$–$C_{35}$) decreasing in order is a classic marine source signature; (2) the high presence of $C_{30}$ hopane compared with $C_{29}$ hopane is indicative of a marine clastic source; (3) a higher abundance of $C_{28}$ bisnorhopane compared with oleanane suggest a higher marine input compared with the above source[58]; (4) a greater abundance of $C_{26}$ tricyclic terpane as compared to $C_{25}$ tricyclic terpane provides evidence of some terrigenous input. Consistent with previous interpretations, we consider this source to be a part of the Wilcox Group oils, more specifically Wilcox Group I as originally described by refs. [59,60].

## Thermal maturity
We adopted standard saturate and aromatic biomarker ratios to develop apparent thermal maturity estimates for each of the oil samples analysed. We discuss these in the context of apparent thermal maturities because we recognize these estimates are based on the bulk weighted average composition of the fluid. The interpretation of the GC/MS analyses of the 12 oils yielded a thermal maturity range of $R_o$ equivalent of 0.9–1.25%, which is classified as main-to-late-stage oil. The interpretation of this thermal maturity range is based on the following: (1) the presence of higher concentrations of high-molecular-weight compared with low-molecular-weight triaromatic steriods, described as % ΔTAS (20–57%); (2) the isomerization of $C_{29}$ ααα−20S relative to $C_{29}$ ααα−20R steranes (0.45–0.48)[61]; (3) the isomerization of 1,2,5-trimethylnapthalene to 1,3,7-trimethylnapthalene represented as ratio TMNr (0.48–0.67) as defined by ref. [62]; (4) the ratio of $C_{23}$ tricyclic terpane to $C_{30}$ hopane, which is calculated as T23/(T23 + H30) (0.10–0.37), where the increased presence of $C_{23}$ tricyclic represents higher maturity.

## Alteration
The use of WOGC and GC/MS analyses help provide a picture of the alteration history of these oils. The shallow Nebo-Hemphill NHH3 fluid is the most biodegraded end member of the samples tested, based on the apparent lack of $n$-alkanes, an API gravity of 21 and the presence of demethylated hopanes. This sample would be classified as severely biodegraded based on the classification of ref. [63]. The remaining Nebo-Hemphill samples have APIs of 32–41 and are classified as unbiodegraded to very slight biodegradation[63]. The Olla Oil Field fluids show evidence of reduced light $n$-alkanes, display similar $n$-alkane-to-isoprenoid ratios and have APIs of 33 to 37, suggesting similar levels of biodegradation between the fluids. Therefore, the Olla fluids would be classified as having very slight biodegradation[63].

## Summary of previous findings
The formation water geochemistry, molecular oil composition and produced gas composition (including $\delta^{13}C$) in the Olla and Nebo-Hemphill oil fields have been previously characterized[9–11]. Microbial methanogenesis was first hypothesised for the Olla Oil Field based on its enriched $\delta^{13}C$ of $CO_2$ (ref. [9]). However, owing to the differences in the isotopic composition between the 2,800-ft sandstone and the other producing formations with the injected material, they concluded that the injected $CO_2$ was not present in the producing zones[10] and consequently any microbial methanogenesis was independent of injection and is probably associated with microbial oxidation of hydrocarbons. The enhanced methanogenesis within the Olla Oil Field compared with other fields produced from the Wilcox Group was thought to be due to the ideal geochemical conditions (that is, suitable range of pH, temperature and salinity, and lack of alternative electron acceptors)[10]. They conclude that the injected $CO_2$ was most likely dissolved and trapped in a gas phase[10]. Further investigation of microbial methanogenesis found that there was little correlation between crude oil biodegradation and methanogenesis; however, the extent of oil biodegradation was correlated with the temperature, salinity and depth of samples[11]. In addition, there has also been an investigation of the microbial community composition within the Olla reservoir[37], which found methanogens present within the system, and no significant difference in microbial communities between the 2,800-ft sandstone and other producing zones[37]. With the addition of the inert noble gas isotopes and clumped isotopes, we are able to re-investigate the fate of $CO_2$ within the Olla reservoirs.

## Formation water chemistry in the Olla Oil Field
The pH and alkalinity of the formation waters within the Olla Oil Field are between 6.5 and 7.5 and between 16.3 meq $kg^{-1}$ and 57.7 meq $kg^{-1}$, respectively[10]. The most dominant anion was $Cl^-$ (1,122–1,621 mmol $l^{-1}$) and the most dominant cation measured was $Na^{2+}$ (1,108–1,630 mmol $l^{-1}$) (Extended Data Table 1), which are typical of basinal brines[64] and in agreement with previous measurements[10]. Other cations present include: $Mg^{2+}$ (119–329 ppm), $Ca^{2+}$ (80–500 ppm), $K^+$ (223–524 ppm), $Ba^{2+}$ (95–165 ppm), $Si^{4+}$ (10.1–12.5 ppm) and $Sr^{2+}$ (77–215 ppm). Measured iron concentrations are less than 0.1 ppm. With the exception of sample O1 where $SO_4^{2-}$ concentrations are 62 ppm, $SO_4^{2-}$ concentrations are less than 2 ppm. Both the iron and $SO_4^{2-}$ concentrations are in agreement with those previously reported[10].

## Identification of injected material
$CO_2$ concentrations in the injected Olla Oil Field (18 ± 12 mol%) are significantly greater than the Nebo-Hemphill Oil Field (1.02 ± 0.69 mol%). $CH_4$ concentrations at Olla are 77 ± 13 mol% (STP) compared with 92.6 ± 8.2 mol% (STP) at Nebo-Hemphill. However, the $\delta^{13}C$ of $CO_2$ of the injected material (0.85 ± 0.91‰) is isotopically distinct from the $\delta^{13}C$ of $CO_2$ now within Olla (13.6 ± 3.2‰) (Fig. 2b). The injected $CO_2$ was sourced from the Black Lake Oil Field, which is located close to the

Sabine Island Complex. The Sabine Island Complex forms part of the Sabine Uplift basement high and has been suggested to be a conduit for mantle-derived fluids[8]. However, the measured $\delta^{13}C$ of $CO_2$ values at Black Lake (0.85 ± 0.91‰)[10] are elevated relative to mantle $CO_2$ (−2‰ to −5‰; refs. [21,22]), probably reflecting some degree of biological modification when the reservoir was shallower and at temperatures favourable to microbial activity. An alternative interpretation for the elevated $\delta^{13}C$ of $CO_2$ at Black Lake is that it is a mixture of mantle-derived and crustal-derived $CO_2$; however, this would increase the $CO_2/^3He$ above that expected in the mantle, which is not observed in the Olla samples.

As the stable isotopes of carbon are frequently modified by chemical or biological processes, the inert nature of noble gases means that they are ideal tracers for injected $CO_2$ in such scenarios. As injectate samples were not collected during $CO_2$-EOR, we do not have access to the noble gas compositions of the material injected into the Olla Oil Field. Nevertheless, it is reasonable that the noble gases record pre-injection conditions at the nearby Nebo-Hemphill Oil Field, and that these are subsequently modified at the Olla Oil Field through mixing of the injected material during the $CO_2$ injection.

The $^4He/^{20}Ne$ ratio is between 3,210 and 20,900 in both fields, which is significantly greater than the atmospheric value of 0.32 and thus there is no significant atmospheric contribution to these samples. The air-corrected He isotopic ratio ($^3He/^4He$), reported relative to the atmospheric ratio ($R_C/R_A$ where air is $1R_A$), in the Olla Oil Field is $1.84 \pm 0.10R_A$ and $0.46 \pm 0.08R_A$ at Nebo-Hemphill (Extended Data Table 2). These are both higher than a typical hydrocarbon system where the majority of He is derived from crustal radiogenic production ($^3He/^4He = 0.02R_A$; ref. [12]), and higher than other fields within the Gulf of Mexico region[8,18,43]. The elevated $^3He/^4He$ values are the result of a resolvable mantle contribution. Using a simple two endmember mixing model, with subcontinental lithospheric mantle ($^3He/^4He = 6.1$; ref. [65]) and crustal endmembers, we resolve an average 7.8% mantle contribution of mantle He in Nebo-Hemphill Oil Field. It is likely that the mantle fluid contribution is coming from the Little Creek collapse structure, which is situated between the two fields[66]. At Olla, the average mantle contribution to $^3He$ is 30%, which suggests that there is an additional source of mantle fluids to this field, which is reasonably explained by the injectate sourced from the Black Lake Oil Field. Ne isotope ratios are elevated at Olla ($^{20}Ne/^{22}Ne$ 10.3 ± 0.2 and $^{21}Ne/^{22}Ne$ 0.0399 ± 0.002) compared with Nebo-Hemphill (9.85 ± 0.09 and 0.0335 ± 0.001, respectively) and air (9.8 and 0.029) (Extended Data Fig. 1, Extended Data Table 3)[67,68,69], again suggesting that the injected material contains mantle-derived fluids.

The $^{40}Ar/^{36}Ar$ values at Nebo-Hemphill are significantly greater than the atmospheric value (298.56; ref. [70]), with measured ratios of 370 ± 3 to 604 ± 5. The Olla Oil Field has a higher $^{40}Ar/^{36}Ar$ (1,476 ± 13 to 2,766 ± 24) (Extended Data Table 3). Elevated $^{40}Ar/^{36}Ar$ ratios can be a result of both an excess of radiogenic $^{40}Ar$ and mantle-derived fluids[12,71]. The difference in the $^{40}Ar/^{36}Ar$ ratios could be due to variable amounts of radiogenic production or fundamental differences in the extent of contact with formation waters between the two fields. However, as all the wells sampled are within the same geological formation, the two fields are in relatively close proximity and have indistinguishable $^4He$ concentrations, it seems unlikely that this would be the cause. Thus, the difference is probably a result of the injected $CO_2$, which probably had mantle-derived $^{40}Ar/^{36}Ar$ excesses, concordant with the observed mantle-derived He and Ne isotopes. Finally, whereas xenon (Xe) isotope ratios at Nebo-Hemphill are indistinguishable from air, the Olla Oil Field showed excesses in some Xe isotope ratios relative to air. For example, the $^{129}Xe/^{130}Xe$ ratio is 6.59 ± 0.01 compared with an atmospheric value of 6.49 (Extended Data Table 3); this excess in $^{129}Xe$ relative to $^{130}Xe$ is consistent with a contribution from mantle fluids with an end member $^{129}Xe/^{130}Xe$ value of 8.2 (ref. [71]). The Olla Oil Field therefore shows a clear mantle component that is not present within the Nebo-Hemphill control field.

The detection of injected fluids within the Olla Oil Field using the conservative noble gas tracers differs from the findings of the previous studies that used the $\delta^{13}C$ of $CO_2$ to conclude that injected $CO_2$ is no longer or never was present in the majority of the sampled wells (for example, ref. [10]). The subsurface behaviour of the injected $CO_2$ within the system can therefore be re-examined using a combination of noble gases and stable isotopes.

## Dissolution within the Olla Oil Field

$CO_2$ is highly soluble in water under reservoir conditions. The most likely trapping mechanisms for free-phase subsurface $CO_2$ are dissolution or precipitation as carbonate minerals mediated by dissolution[4,21]. A decrease in the $CO_2/^3He$ correlates with increasing $^4He$ and $^{20}Ne$ concentrations within the Olla Oil Field (Extended Data Fig. 2). $^{20}Ne$ (and often $^4He$)[4] are derived from the formation water, suggesting that $CO_2$ trapping in the Olla Oil Field is also proportional to the amount of formation water that has been contacted and degassed, consistent with dissolution within the field[4]. Dissolution (solubility trapping) has previously been found to be a major sink for $CO_2$ (ref. [4]) and imparts a slight pH-dependent fractionation towards a light isotopic composition[21]. However, this fractionation effect is marginal compared with the amount of $CO_2$ that can be trapped and the fractionation that can be imparted by methanogenesis–AOM cycling (Fig. 2a).

Given the presence of discontinuous coal beds within the Wilcox Group[57,72], it is possible that some of the injected $CO_2$ may have been sorbed onto these coal beds as $CO_2$ has a much greater affinity to coal than $CH_4$ (ref. [9]). The resulting isotopic fractionation imparted by this process remains poorly constrained, with some studies suggesting a greater than 5‰ shift[73,74] whereas others suggest negligible effects[75–77] on the $\delta^{13}C$ of $CO_2$. Coal sorption would cause fractionation in the opposition direction of the $CH_4$ isotopic signature. Given the heavier $\delta^{13}C$ of $CO_2$ and $\delta^{13}C$ of $CH_4$ isotopic signatures in the injected Olla Oil Field, it is unlikely that $CO_2$ sorption to coal is a major $CO_2$ sink in the field, in agreement with previous findings[10].

## Subsurface behaviour of carbon at Olla

The Olla samples are enriched in both the $\delta^{13}C$ of $CO_2$ and the $\delta^{13}C$ of $CH_4$ compared with Nebo-Hemphill and the injected $CO_2$ values. The offset between the $\delta^{13}C$ of $CH_4$ and the $\delta^{13}C$ of $CO_2$ in the Olla Oil Field are close to those expected under thermally driven isotope exchange equilibrium for the observed temperatures (Fig. 2b). Carbon isotopes of $CO_2$ and $CH_4$ approaching equilibrium in hydrocarbon systems with current reservoir conditions have previously been observed[24]. However, the thermal equilibrium of $CO_2$ and $CH_4$ is kinetically limited at the current temperatures (29.2–50.7 °C) in the Olla reservoirs and would not occur on the timescale of injection, without a process to enhance the reaction rate[78]. A combination of both microbial methanogenesis and AOM active within the field would allow for the system to be approaching equilibrium under the current thermal conditions within the reservoir. Further observations detailed below provide evidence for this conclusion and allow for the net microbial methanogenesis to be quantified.

The clumped isotopologues of $CH_4$ ($\Delta^{13}CH_3D$ and $\Delta^{12}CH_2D_2$) can provide independent constraints on its origin within petroleum systems[25–29]. Thermogenic $CH_4$ is expected to have a composition consistent with the source-rock maturation temperature, which is typically at or above 157 °C (ref. [25]). At Olla, we expect $CH_4$ generation to have occurred at 163 ± 18 °C (based on biomarker maturity estimates following ref. [35]). Microbial $CH_4$ is variable in its clumped isotopic compositions with both non-equilibrium[28,29] and apparent equilibrium signatures reported in samples obtained from the deep subsurface[26,29]. Microbial methanogenesis forms $CH_4$ characterized by low $\Delta^{13}CH_3D$ and extremely low $\Delta^{12}CH_2D_2$ (refs. [28,29]), whereas AOM tends to drive the isotopologue composition of $CH_4$ towards equilibrium, with equilibrium expected to be reached at a faster rate in $\Delta^{13}CH_3D$ compared

with in $\Delta^{12}CH_2D_2$ (refs. [29,30,34,79–81]). In the Olla Oil Field, we see $\Delta^{13}CH_3D$ of 3.45–5.62‰ (apparent temperature of $29^{+14}_{-12}$ °C to $128^{+17}_{-16}$ °C) and $\Delta^{12}CH_2D_2$ of 9.13–12.4‰ (apparent temperature of $87^{+13}_{-11}$–$132^{+30}_{-23}$ °C) (Fig. 3, Extended Data Table 2); these temperatures preclude a thermogenic origin for the $CH_4$ and are instead consistent with a microbial contribution. Samples O4 and O6 are in internal clumped isotopologue equilibrium at temperatures lower than expected for the thermogenic $CH_4$ and appear to be approaching equilibrium with current reservoir conditions. The remaining samples from Olla have already reached apparent equilibrium with the current reservoir conditions in $\Delta^{13}CH_3D$ but have yet to reach apparent equilibrium in $\Delta^{12}CH_2D_2$ probably due to kinetic effects. The clumped isotopologues at Olla are approaching equilibrium at reservoir temperatures, which would be accounted for by active AOM within the Olla Oil Field. This supports the conclusion that there is active microbial methanogenesis and AOM driving the carbon isotopes to approach equilibrium under current thermal conditions in the Olla reservoirs.

There is microbial degradation of the higher alkane gases (Extended Data Fig. 3) within the Olla Oil Field. Propane is preferentially consumed during microbial degradation and thus in biodegraded gases, propane will see the most pronounced $\delta^{13}C$ enrichment[36]. This $\delta^{13}C$ enrichment is clear within the Olla gases (Extended Data Table 3) and supports active microbial hydrocarbon degradation within the field.

The presence of methanogenic activity and the idea of a methanogenic 'hotspot' in the Olla reservoir has been previously suggested[9,10]. The environmental conditions for which methanogenesis can occur are relatively well defined. Within nutrient-poor environments such as oil reservoirs or coal beds, methanogens can survive at temperatures up to 80 °C, with the largest growth rates and thus greatest amount of methanogenic activity between 40 °C and 50 °C (refs. [36,82–84]). Methanogenic activity is greatest at near neutral pH and is inhibited at pH <4 and >9 (ref. [84]). Formation water salinity can also inhibit methanogenesis above $Cl^-$ concentrations of 1,500 mM, with greatest growth between 200 nM and 600 mM (refs. [82,83,85]). The presence of ferric iron and sulfate at concentrations greater than 1 mM can also limit methanogenesis (as a result of the dominance of other bacteria)[86–89]. Notably, the Olla Oil Field exhibits ideal environmental conditions for methanogenesis, in terms of pH (6.5–7.5)[10], temperature (29.2–50.7 °C), salinity (1,094–1,622 mM), sulfate (<0.6 mM $l^{-1}$)[10] and iron concentrations (≤0.1 mM $l^{-1}$) (Methods).

In addition, geochemical and isotopic parameters associated with microbial methanogenesis in organic-rich settings are also well established[10,11,83,88,90]. Traditionally, carbon isotopes of $CH_4$ have been used to determine the presence of biogenic $CH_4$ (typically less than −60‰)[91–93] (Extended Data Fig. 3a). However, methanogenesis can result in heavy biogenic $CH_4$ as the residual $CO_2$ pool gets increasingly elevated, consistent with observations at Olla (Fig. 2b). In addition, where methanogenesis has occurred, formation water alkalinity is greater than 10 meq $kg^{-1}$, the acetate concentrations are lower than 1 mM, the Ca/Mg ratios are less than 1.5 and the $\delta^{13}C$-DIC is greater than +20‰ (refs. [87,88,90]). Within the Olla Oil Field, the alkalinity ranges between 21.4 meq $kg^{-1}$ and 57.7 meq $kg^{-1}$, the average Ca/Mg ratio is 0.2 and the $\delta^{13}C$-DIC is 14.2–28.7‰, again indicating that methanogenesis has occurred within the field.

SSU rRNA gene-sequencing analysis of the microbial communities within the Olla Oil Field (samples O1 and O4) indicate that within the archaeal sequences, hydrogenotrophic methanogens, methanol and methylotrophic methanogens, and ANME assigned sequences are present (Extended Data Fig. 5). This microbial array would facilitate both microbial methanogenesis and AOM within the field.

During the cycling of carbon between methanogenesis and AOM, we would not expect a change in the isotopic composition of hydrogen. It is possible that as a result of microbial activity or the reaction kinetics that extracellular hydrogen is added to the system. This hydrogen could be sourced from the water and free hydrogen from the degradation of the higher hydrocarbon gases or from the oil. As a result, we are unable

to determine whether the hydrogen isotopes within the system are also approaching equilibrium under current reservoir temperatures.

It is possible that the $CO_2$ and $CH_4$ isotopes in the Olla system are reflecting purely the kinetic process of methanogenesis. The difference in the $\delta^{13}C$ of $CO_2$ and the $\delta^{13}C$ of $CH_4$ at Olla (53.7–64.4‰) is consistent with that expected from microbial methanogenesis (about 58‰; ref. [94]). In such a scenario, all the $CH_4$ at Olla would be microbial in origin, thus requiring any thermogenic $CH_4$ component originally in the system to be overwhelmed. The high concentration and mixed thermogenic–microbial composition of $CH_4$ within the Nebo-Hemphill Oil Field (Fig. 3, Methods) is inconsistent with such a scenario. In addition, the clumped isotopologues at Olla show a clear need for AOM within the field and 16S rRNA gene sequencing shows active methanotrophs, which is not compatible with a scenario of purely microbial methanogenesis dominating the system. Therefore, future studies investigating the processes affecting $CO_2$ in potential CCS storage targets will benefit from a complete geological, geochemical and microbiological characterization of the field study site.

## Subsurface behaviour of carbon in the Nebo-Hemphill Oil Field

The carbon isotopic composition of $CO_2$ and $CH_4$ at Nebo-Hemphill is depleted compared with the Olla Oil Field (Fig. 2b). Despite this, the difference in the $\delta^{13}C$ of $CO_2$ and the $\delta^{13}C$ of $CH_4$ at Nebo-Hemphill overlaps with that of Olla (50.8–68.1‰ and 53.7–64.4‰, respectively). The degree of isotopic fractionation at Nebo-Hemphill is consistent with thermal equilibrium at 29.8–85.3 °C, in agreement with present-day reservoir temperatures (29.7–57.1 °C). In addition, the $\delta^{13}C$ of $CO_2$ at Nebo-Hemphill is enriched compared with that typically expected for crustal or mantle-sourced $CO_2$ (less than 2‰)[21,22], consistent with a small amount of microbial methanogenesis within the field. We propose that there is also microbial methanogenesis (in agreement with previous studies[10]) and AOM within the Nebo-Hemphill Oil Field; however, unlike at Olla, this has not been enhanced by any injection to the field.

Microbiological analysis of NHH3 and NHH6 show that hydrogenotrophic methanogens, methanol and methylotrophic methanogens, and ANME assigned sequences are present within Nebo-Hemphill, in agreement with both active methanogenesis and AOM within the field (Extended Data Fig. 5). Clumped isotopologues of $CH_4$ within the Nebo-Hemphill Oil Field give apparent temperatures for $\Delta^{13}CH_3D$ of $78^{+14}_{-12}$–$166^{+34}_{-29}$ °C and apparent temperatures for $\Delta^{12}CH_2D_2$ that are greater than expected for thermogenic $CH_4$ in the field (163 ± 18 °C, Fig. 3). They suggest that the $CH_4$ is likely to be dominantly thermogenic with some microbial contributions. Samples NHH1, NHH2 and NHH8 have apparent temperatures below that expected for thermogenic $CH_4$ and appear to be approaching low-temperature equilibrium in $\Delta^{13}CH_3D$, consistent with active methanogenesis and AOM cycling within the field.

## Modelling of the Pannonian Basin

The unequivocal identification and quantification of $CO_2$-related microbial methanogenesis at Olla has globally significant implications, and warrants the reconsideration of reservoir processes at other sites worldwide. Re-examination of previously published datasets allows the elucidation of the significance of methanogenesis in different reservoir environments, especially in systems that have or had naturally high $CO_2$ concentrations, such as the Pannonian Basin. A significant proportion of the $CO_2$ found in natural gas fields within the Pannonian Basin is of mantle origin[20]. The initial, $CO_2/^3He$ in the Szegholm North, Szegholm South, Ebes and Hajduszoboszlo gas fields was estimated to be $7.9 ± 5.4 × 10^9$, consistent with the European subcontinental lithospheric mantle range ($0.6–40 × 10^9$) and the mid-ocean ridge basalt mantle range ($2 ± 1 × 10^9$)[95–97] (Extended Data Fig. 7a). Since emplacement, significant $CO_2$ trapping in many fields was identified from lower observed $CO_2/^3He$ ratios (for example, the Szegholm North and South, Ebes and Hajduszoboszlo gas fields) and speculation that some of the

$CO_2$ may have been converted into $CH_4$ (ref. [20]), although no viable mechanism was identified.

Using an average geothermal gradient of 45 °C km$^{-1}$ the temperature within these fields is predicted to be between 40 °C and 105 °C (with temperatures being up to 20 °C cooler at the time of $CO_2$ emplacement)[98], and thus within the range of microbially driven methanogenesis. In addition, the presence of thermogenic hydrocarbons would also ensure $H_2$ availability, similar to the Olla and Nebo-Hemphill oil fields. From our work in Olla, we surmise that the Pannonian Basin may be analogous to Olla and that hydrogenotrophic microbial methanogenesis active within the field may provide the missing mechanism for $CO_2$ removal, which was previously unidentified, and point to this mechanism as an important subsurface process operating on systems naturally enriched in $CO_2$ as well as anthropogenically $CO_2$ injected oil and gas fields.

In the absence of supporting clumped isotopologue data, if we assume that the system is methanogenesis driven we can nevertheless construct similar methanogenesis–dissolution models using published data from the Szegholm South Gas Field (where multiple $CO_2$ isotope data are available). By comparing the observed $CO_2/^3$He ratios against the $CO_2$ concentration, a clear correlation emerges (Extended Data Fig. 7a). Attributing this to the effect of dissolution (a pre-requisite for microbial methanogenesis), it is possible to extrapolate back to an initial $CO_2/^3$He end member value of $13.3 \times 10^9$, consistent with previous estimates.

Our model assumes mantle-derived $CO_2$ in the field with a typical isotopic composition of $\delta^{13}$C of $CO_2 = -5$‰ (Extended Data Fig. 7b). Taking this as a starting point, $16.4 \pm 0.8$ times more dissolution than methanogenesis is required in the Szegholm South Gas Field to produce the measured $\delta^{13}$C of $CO_2$. As such, proportionally less methanogenesis, relative to the Olla Oil Field, would be expected here as the gas fields are not being constantly perturbed via water injection and therefore nutrient availability is probably much more limited. The presence of biogenic $CH_4$ (through $CO_2$ conversion) can also be clearly identified from the measured $C_1/C_N$ in the Szegholm South Gas Field ($0.946 \pm 0.004$), which is greater than what would be expected in a purely thermogenic system ($0.909$). Using the excess in the $C_1/C_N$ ratios, we determine that $41 \pm 5\%$ of the $CH_4$ currently present in the Szegholm South Gas Field part of the basin is biogenic. Similar $C_1/C_N$ excesses are seen in the other gas fields in the Pannonian Basin[99]. In addition, other fields such as Szegholm North[20,99], Ebes[20,99], Hajduszoboszlo[20,99], Kismarja[20,99], Répcelak[100] and Mihàly[100] (Pannonian Basin), Subei[101] (China) and JM Brown Bassett[95] (Permian Basin) also show geochemical signatures consistent with the occurrence of methanogenesis. Thus, we conclude that although methanogenesis is not the dominant $CO_2$ loss mechanism, it may play a substantial role that has not previously been considered in many natural systems.

## Data availability

The geochemical data that support the findings of this study are available in the NERC EDS National Geoscience Data Centre at https://doi.org/10.5285/a4070f5d-2064-4caf-a82c-79a786d6af9e. The microbial SRA and biosample data can be found at https://www.ncbi.nlm.nih.gov/bioproject/PRJNA744568. Source data are provided with this paper.

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

**Acknowledgements** R.L.T. was supported by a Natural Environment Research Council studentship (grant reference NE/L002612/1). C.J.B. and P.H.B. acknowledge A. Regberg and B. Meurer for their support of the project and help with sample collection. C.J.B. was part supported by an Earth4D CIFAR fellowship. P.H.B. was supported by NSF awards 1923915 and 2015789. O.W. was supported by Natural Sciences and Engineering Research Council of Canada Discovery and Accelerator grants awarded to the Sherwood Lollar research group and acknowledges B. Sherwood Lollar's support for the project. Z.M.S. acknowledges J. Biddle and G. Christman for their help in generating the microbial data.

**Author contributions** The project was conceived by R.L.T., P.H.B., M.L. and C.J.B. R.L.T. performed the noble gas isotopic analysis, managed the project and prepared the first draft of the manuscript. P.H.B. collected the samples. Z.M.S., H.X. and B.S. conducted the microbial, clumped isotope and molecular geochemical analysis, respectively. Clumped isotope modelling was developed by H.X. and J.M.E. Integrated noble gas and stable isotope modelling was developed by R.L.T., P.H.B., M.L., C.J.B., O.W. and D.J.B. All authors contributed to the final manuscript.

**Competing interests** M.F. and B.S. are employed by/an employee of ExxonMobil Upstream Integrated Solutions Company. Z.M.S. is employed by/an employee of ExxonMobil Research and Engineering Company. When the manuscript was first submitted, M.L. was employed by/an employee of ExxonMobil Upstream Integrated Solutions Company. M.L. is now employed by/an employee of Aker BP. The views expressed are those of the author(s) and not necessarily those of ExxonMobil Upstream Integrated Solutions Company, ExxonMobil Research and Engineering Company, or Aker BP.

**Additional information**
**Correspondence and requests for materials** should be addressed to R. L. Tyne, P. H. Barry, M. Lawson or C. J. Ballentine.

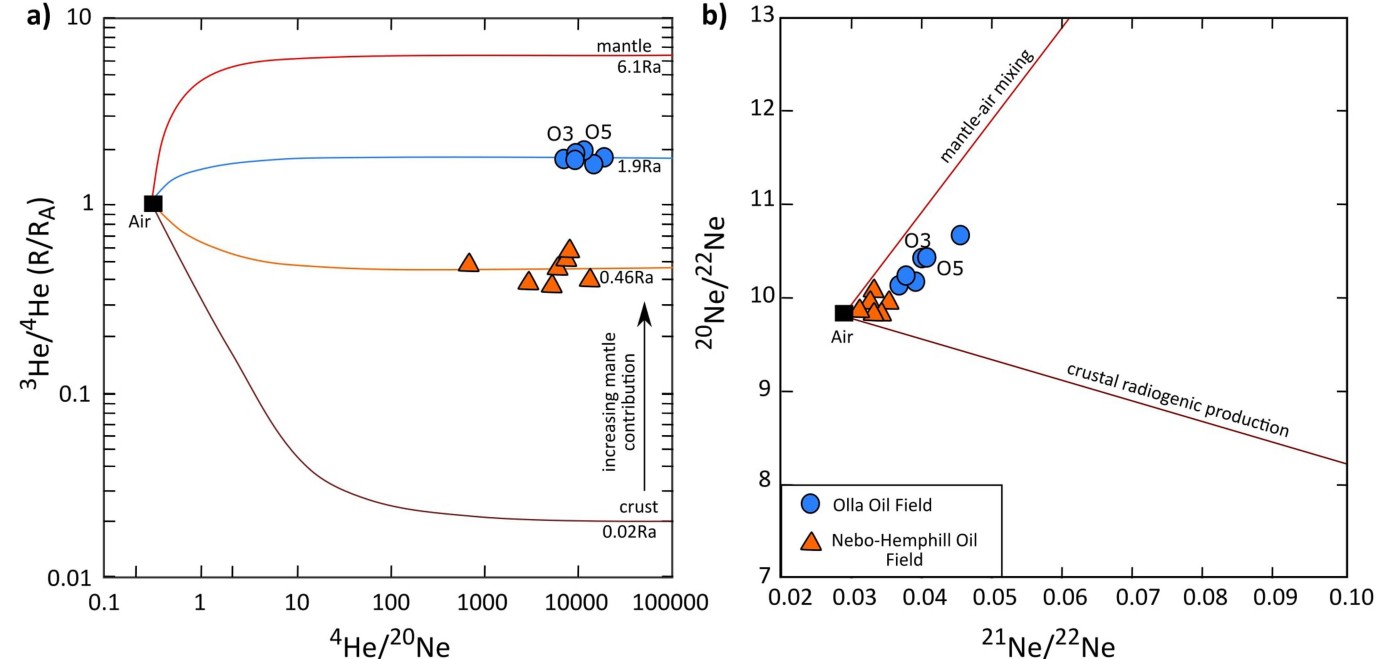

**Extended Data Fig. 1 | Relationship between noble gas isotopic ratios in the Olla (blue circles) and Nebo-Hemphill (orange triangles) oil fields.** Both fields show a resolvable mantle component that is significantly greater in the Olla Oil Field. One standard deviation error bars are within symbol size. **a**, He isotopic ratios versus $^4$He/$^{20}$Ne ratio. Air-corrected He isotopes are reported relative to air ($R_A = 1.38 \times 10^{-6}$ (ref. [69])). All samples are in excess of typical radiogenic production (0.02$R_A$), requiring a mantle contribution. High $^4$He/$^{20}$Ne

indicates negligible air contamination in the samples. **b**, Three neon isotope plot. The crustal production line[68] reflects the observed trend in typical crustal fluids. The mantle-air line represents mixing between air and a MORB-like endmember[67]. Data are consistent with mixing between air and a mantle-rich component. In both He and Ne isotopes, the mantle signal is stronger in the Olla Oil Field.

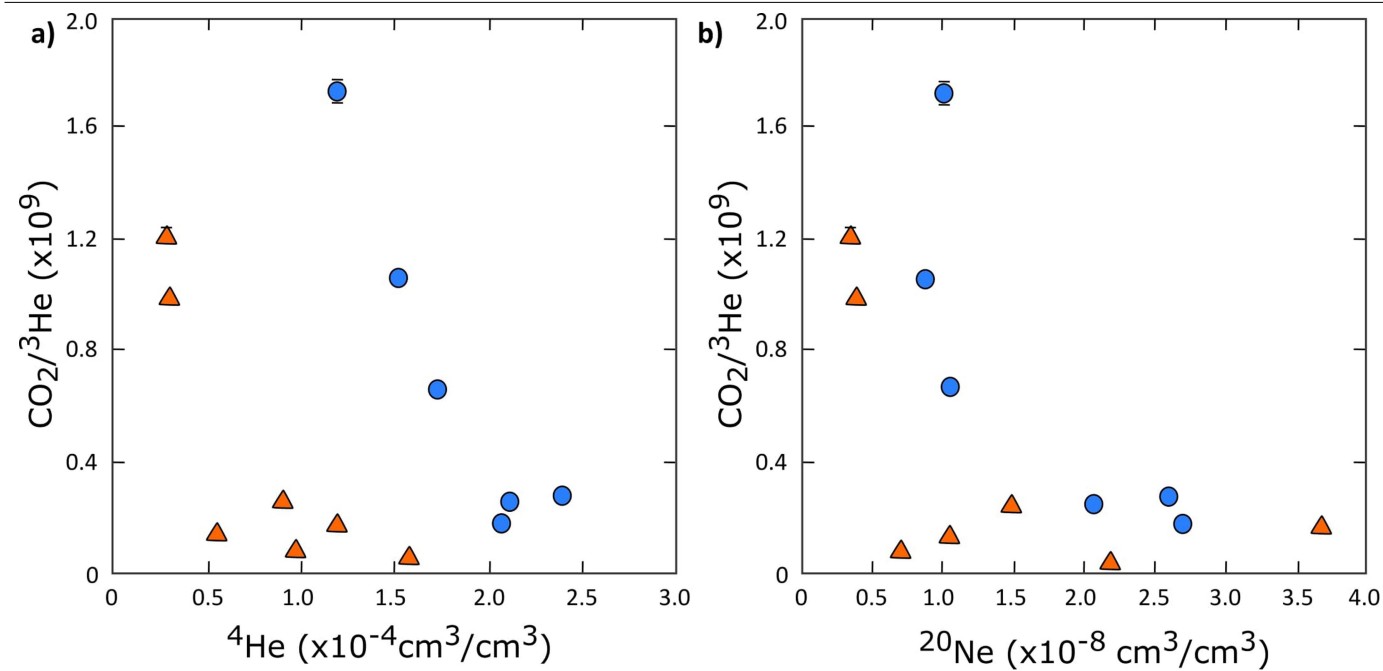

**Extended Data Fig. 2 | CO₂/³He variation vs. a, b,** ⁴He (**a**) and ²⁰Ne (**b**) abundances in the Olla (blue circles) and Nebo-Hemphill (orange triangles) oil fields. $1\sigma$ level of uncertainty is shown on the measured samples. A negative correlation between ⁴He, which accumulates in formation waters as a function of time and ²⁰Ne (which is groundwater derived) and CO₂/³He, has been interpreted in other CO₂-rich natural gas fields[4] to support the importance of the formation water in controlling CO₂ removal.

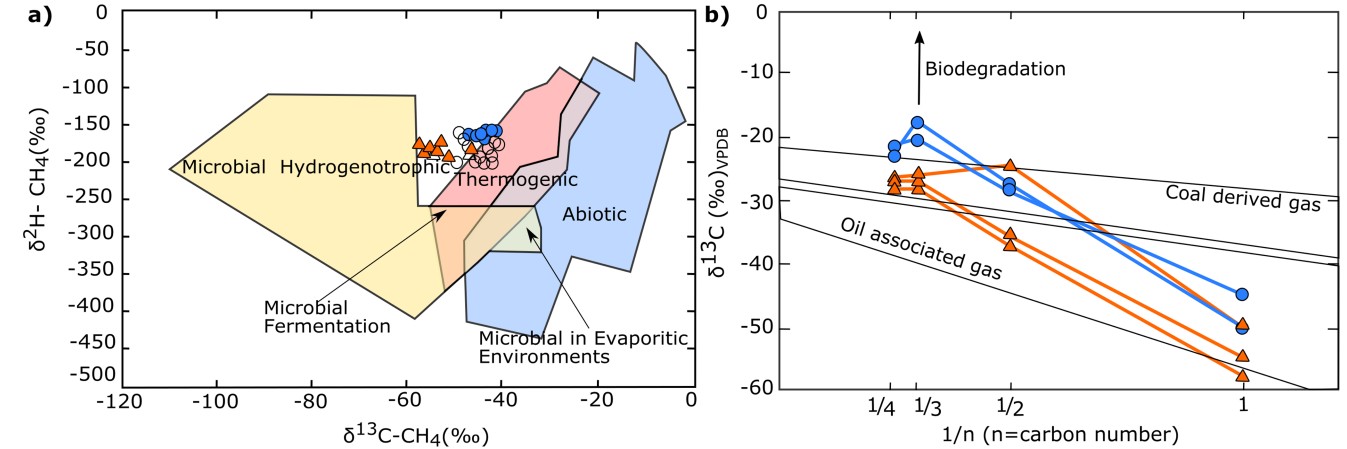

**Extended Data Fig. 3 | Hydrocarbon isotopic composition in the Olla (blue circles) and Nebo-Hemphill (orange triangles) oil fields.**
**a**, Comparison of $\delta^{13}C$-$CH_4$ and $\delta D$-$CH_4$, with typical $CH_4$ provenances after Whiticar[93]. One standard deviation error bars are within symbol size. In isolation, the Olla samples (blue circles, this study; unfilled circles, from ref.[10]) are consistent with a mainly thermogenic origin, whereas Nebo-Hemphill (orange triangles, this study; unfilled triangles, from ref.[10]) appears to be a microbial-thermogenic mix. Clumped isotopologues nevertheless provide clear evidence that the Olla $CH_4$ is microbial in origin (Fig. 3). **b**, Plot of $\delta^{13}C$ of $C_n$ versus $1/C_n$ variation in natural gases[92]. One standard deviation error bars are within symbol size. Both the Olla (blue circles) and Nebo-Hemphill (orange triangles) fields show an enrichment in their isotopic signature especially in propane; however, this is most pronounced within the Olla Oil Field.

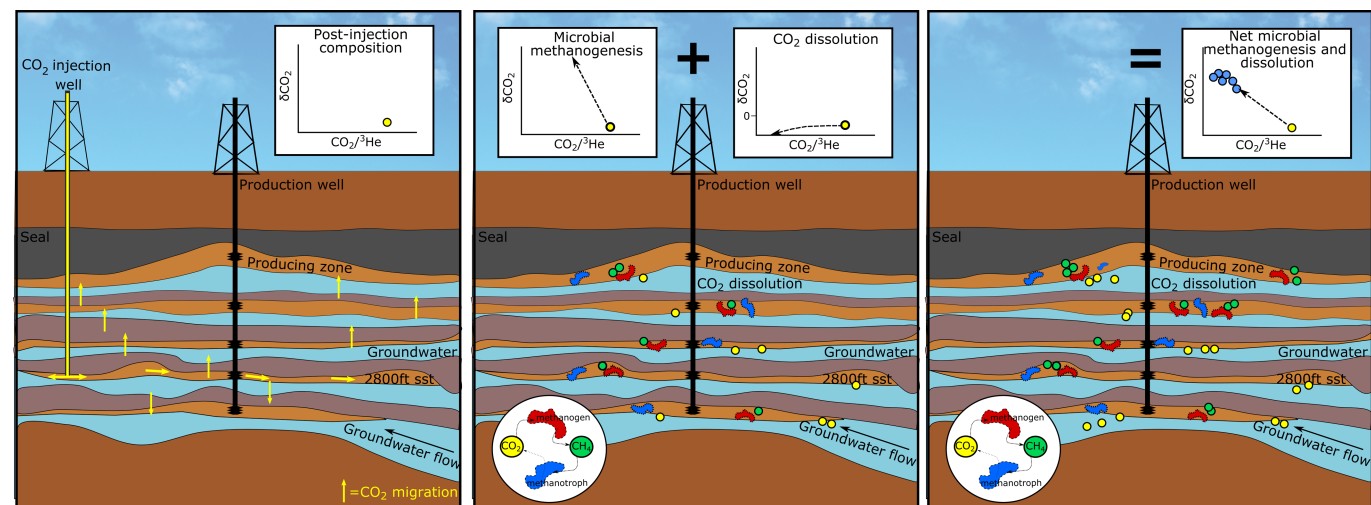

**Extended Data Fig. 4 | Schematic of the processes occurring within the Olla Oil Field and resulting changes in the isotopic composition within the field.** The first panel shows $CO_2$ injection into the 2,800-ft sandstone (sst) and subsequent lateral and vertical migration (yellow arrows) to all the producing formations within the field. The middle panel demonstrate the different processes (microbial methangonesis, AOM and dissolution), which are occurring in the field and their effects on the isotopic composition. Yellow circles represent $CO_2$ dissolution and green circles are the net $CH_4$ production from microbial activity. Hydrogen for microbial methanogenesis is sourced from the hydrocarbons and water. The final panel shows the current state of the reservoir.

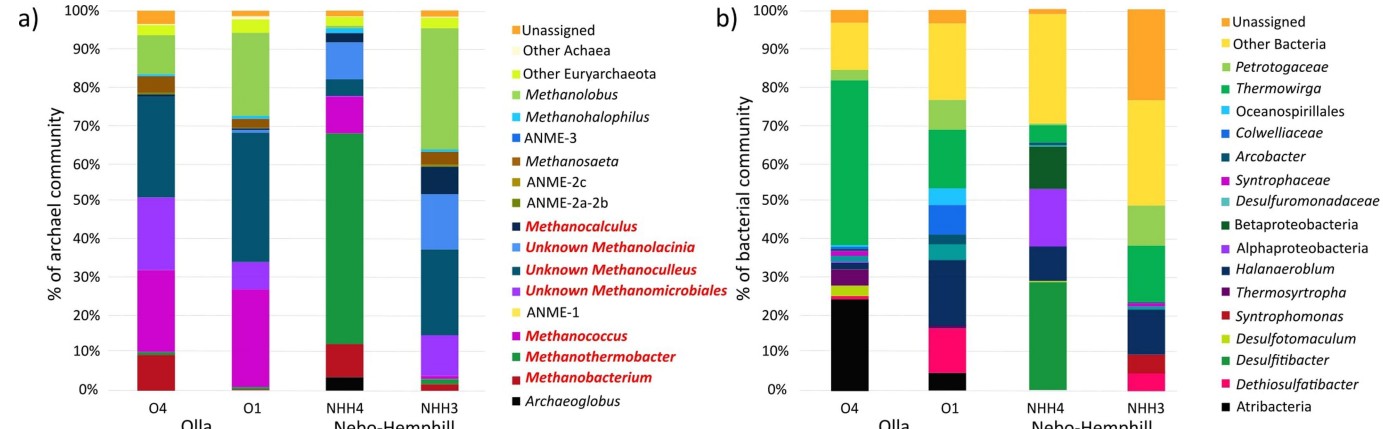

**Extended Data Fig. 5 | Microbial community analysis of most abundant taxa from the SSU rRNA gene-sequence region.** Bar charts represent the most resolved taxa from amplicon sequence variants generated in QIIME 2[49,50,51] for archaeal primers (**a**) and bacterial primers (**b**). Hydrogenotrophic methanogenesis-capable taxa are represented with bold red font.

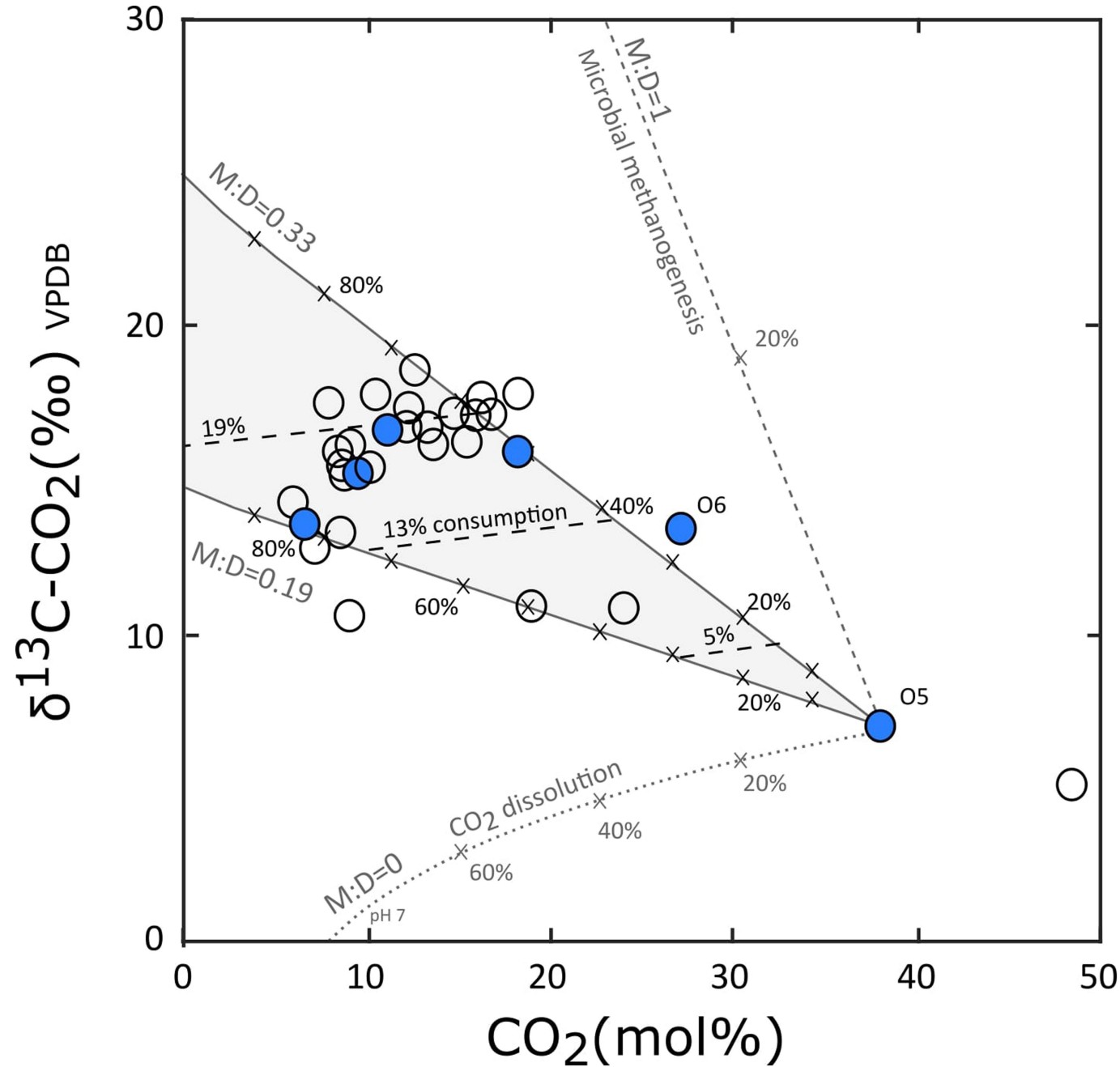

**Extended Data Fig. 6 | Carbon isotopic composition of the Olla samples as a function of $CO_2$ concentration for data from this study (dark blue circles) and those collected without $^3$He analysis[10] (unfilled circles).** One standard deviation error bars are within symbol size. Dashed lines show endmember methanogenesis and dissolution fractionation (at pH 7) trajectories. Tick marks represent the total amount of $CO_2$ trapping within the system, relative to sample O5. The shaded region represents trapping by the combination of both microbial methanogenesis and dissolution. The same combination of dissolution and microbial methanogenesis has been modelled (grey region, cf. Fig. 2). Lines labelled 'consumption' at 5%, 13% and 19% show the proportion of original $CO_2$ that has been removed by net microbial methanogenesis. One sample from ref.[10] has a $CO_2$ concentration greater than our 'most pristine' $CO_2/^3$He sample (O5). This illustrates that by using sample O5 as our least altered composition we calculate a conservative (minimum) amount of $CO_2$ consumption, and that initial $CO_2$ concentrations were probably greater.

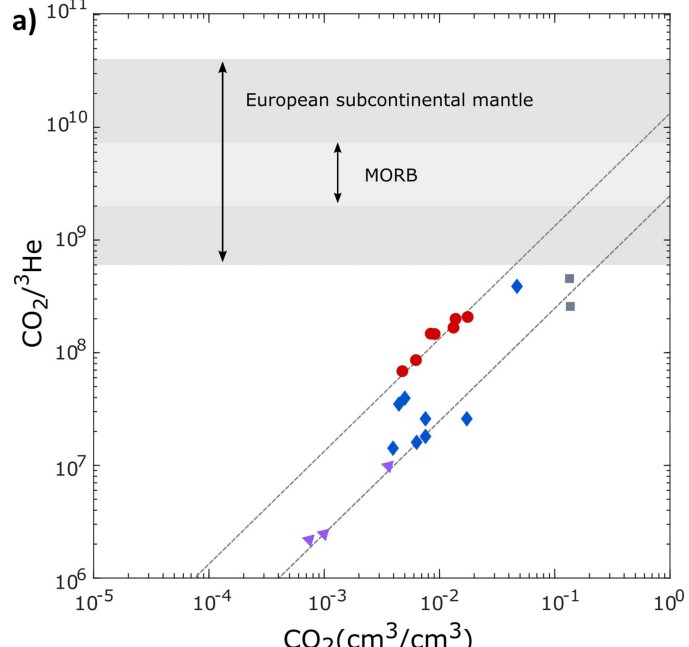

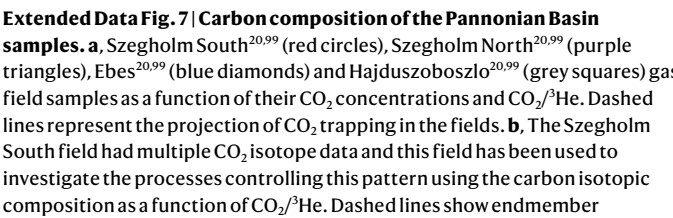

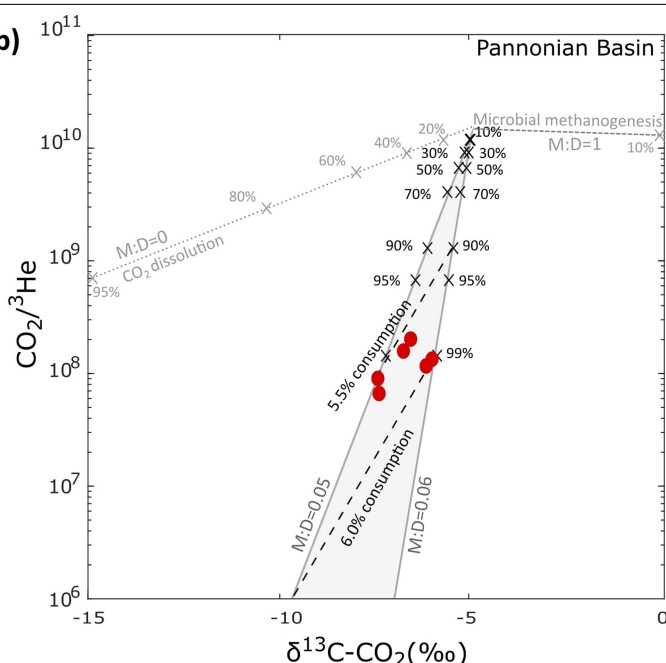

**Extended Data Fig. 7 | Carbon composition of the Pannonian Basin samples. a**, Szegholm South[20,99] (red circles), Szegholm North[20,99] (purple triangles), Ebes[20,99] (blue diamonds) and Hajduszoboszlo[20,99] (grey squares) gas field samples as a function of their $CO_2$ concentrations and $CO_2/{}^3He$. Dashed lines represent the projection of $CO_2$ trapping in the fields. **b**, The Szegholm South field had multiple $CO_2$ isotope data and this field has been used to investigate the processes controlling this pattern using the carbon isotopic composition as a function of $CO_2/{}^3He$. Dashed lines show endmember methanogenesis and dissolution fractionation (at pH 7) trajectories. Tick marks represent the total amount of $CO_2$ trapping within the system. The shaded region represents trapping by the combination of both microbial methanogenesis and dissolution. The upper and lower methanogenesis: dissolution ratios (M:D) are 0.050 and 0.060, respectively, showing dissolution accounts for approximately 16 times (M:D = 0.055) more $CO_2$ removal than microbial methanogenesis. Lines labelled 'consumption' show the proportion of original $CO_2$ that has been removed by net microbial methanogenesis.

**Extended Data Table 1 | Geochemistry and isotopic composition of the formation waters**

| Sample | Latitude, Longitude | Producing zone | Depth m | Cl mmol/l | Na mmol/l | Mg ppm | Ca ppm | K ppm | Ba ppm | Si ppm | Sr ppm | Fe ppm | SO$_4^{2-}$ ppm | Bicarbonate ppm |
|---|---|---|---|---|---|---|---|---|---|---|---|---|---|---|
| O1 | 31.83128, -92.2095 | L Cent | 677 | 1435 | 1409 | 222 | 181 | 223 | 26.7 | 11.3 | 83 | N.D. | 62 | 3655 |
| O2 | 31.83015, -92.2093 | L Cent | 656 | | | | | | | | | | | |
| O3 | 31.82453, -92.2137 | 2800 ft | 835 | | | | | | | | | | | |
| O4 | 31.80899, -92.2297 | U Cent | 663 | 1122 | 1108 | 329 | 80.2 | 228 | 95 | 10.1 | 77 | N.D. | <1 | 3912 |
| O5 | 31.80119, -92.2224 | 2800 ft | 842 | 1357 | 1340 | 305 | 500 | 420 | 165 | 10.6 | 215 | N.D. | <1 | 3164 |
| O6 | 31.76926, -92.2133 | H2-SD | 1051 | 1621 | 1630 | 119 | 343 | 524 | 158 | 12.5 | 207 | N.D. | 2 | 3486 |
| NHH1 | 31.60037, -92.1668 | Wilcox B2 | 1211 | 1538 | 1502 | 581 | 761 | 212 | 198 | 12.0 | 265 | N.D. | <1 | 812 |
| NHH2 | 31.59793, -92.1468 | Wilcox A2 | 1027 | | | | | | | | | | | |
| NHH3 | 31.58830, -92.1752 | T Wilcox | 753 | 1497 | 1452 | 495 | 924 | 157 | 188 | 7.2 | 267 | 2.1 | <1 | 108 |
| NHH4 | 31.58895, -92.1775 | Wilcox C3 | 1099 | 1579 | 1507 | 548 | 862 | 188 | 210 | 10.4 | 266 | 2.3 | <1 | 308 |
| NHH5 | 31.58907, -92.1666 | Wilcox F3a & F3b | 1244 | | | | | | | | | | | |
| NHH6 | 31.58238, -92.1487 | Wilcox F2 | 1202 | 1633 | 1562 | 474 | 1122 | 195 | 216 | 10.4 | 270 | 7.5 | <1 | 38 |
| NHH7 | 31.57380, -92.1512 | Wilcox H2 | 1273 | 1786 | 1662 | 481 | 1618 | 242 | 203 | 10.4 | 324 | 20.2 | <1 | 162 |
| NHH8 | 31.56586, -92.1557 | Wilcox A1 | 1023 | | | | | | | | | | | |

**Extended Data Table 2 | Sample location, reservoir temperature, major gas species, carbon isotopes and clumped isotope data**

| Sample | Reservoir temp* | CO₂† | CH₄† | δ¹³C-CO₂ | δ¹³C-CH₄ | δ¹³C-CO₂ – CH₄ | CO₂ /³He | ³He/⁴He | ⁴He | Δ¹³CH₃D | Apparent Δ¹³CH₃D Temperature | Δ¹²CH₂D₂ | Apparent Δ¹²CH₂D₂ Temperature |
|---|---|---|---|---|---|---|---|---|---|---|---|---|---|
| | $°C$ | mol %(STP) | mol %(STP) | ‰ VDPB | ‰ VDPB | ‰ VDPB | x10⁸ | | x10⁻⁶ cm³/cm³(STP) | | $°C$ | | $°C$ |
| O1 | 39.1 | 11 | 86 | 16.4 | -47.6 | 64 | 2.77 ± 0.06 | 1.78 ± 0.04 | 23.7 ± 0.2 | 4.89 ± 0.36 | $56^{+17}_{-15}$ | 12.41 ± 1.29 | $88^{+15}_{-14}$ |
| O2 | 39.7 | 6.7 | 89 | 13.7 | -49.3 | 63 | 1.92 ± 0.04 | 1.79 ± 0.04 | 20.7 ± 0.2 | 4.47 ± 0.34‡ | $74^{+18}_{-14}$ | | |
| O3 | 44.4 | 9.4 | 88 | 15.1 | -49.3 | 64.4 | 2.47 ± 0.06 | 1.93 ± 0.04 | 20.9 ± 0.2 | 5.62 ± 0.31‡ | $30^{+10}_{-10}$ | 9.13 ± 1.60 | $132^{+30}_{-23}$ |
| O4 | 39.4 | 18 | 73 | 15.8 | -48.2 | 64 | 6.60 ± 0.15 | 1.72 ± 0.04 | 16.0 ± 0.2 | 3.93 ± 0.52 | $100^{+29}_{-25}$ | 12.19 ± 1.39 | $90^{+17}_{-15}$ |
| O5 | 44.5 | 38 | 57 | 7.05 | -46.7 | 53.75 | 17.3 ± 0.4 | 1.98± 0.04 | 11.8 ± 0.1 | 5.66 ± 0.39 | $29^{+14}_{-12}$ | 12.46 ± 1.07 | $87^{+13}_{-11}$ |
| O6 | 50.7 | 27 | 67 | 13.3 | -44.8 | 58.1 | 10.6 ± 0.2 | 1.81 ± 0.04 | 15.1 ± 0.2 | 3.45 ± 0.26‡ | $128^{+17}_{-16}$ | 9.82 ± 1.53 | $121^{+25}_{-21}$ |
| NHH1 | 51 | 1.9 | 97 | 13.9 | -46 | 59.9 | | | | 3.77 ± 0.37 | $109^{+34}_{-31}$ | 4.27 ± 1.29 | $252^{+84}_{-44}$ |
| NHH2 | 50 | 0.28 | 96 | 0.33 | -49.8 | 49.5 | 1.44 ±0.06 | 0.38 ± 0.04 | 5.41 ± 0.04 | 4.38 ± 0.28‡ | $78^{+13}_{-12}$ | 3.85 ± 1.81 | $270^{+118}_{-65}$ |
| NHH3 | 41.9 | 0.77 | 98 | 7.1 | -61 | 68.1 | 1.79 ± 0.05 | 0.39 ± 0.01 | 11.8 ± 0.1 | 3.17 ± 0.35 | $146^{+22}_{-20}$ | 5.03 ± 1.24 | $224^{+48}_{-36}$ |
| NHH4 | 52.1 | 1.7 | 94 | 9.4 | -54.5 | 63.9 | 9.97 ± 0.12 | 0.52 ± 0.01 | 2.92 ± 0.03 | 3.16 ± 0.54 | $147^{+32}_{-39}$ | 1.56 ± 1.67 | $443^{+n/a}_{-142}$ |
| NHH5 | 56 | 1.8 | 76 | 3.08 | -57 | 60.08 | 12.2 ±0.3 | 0.58 ± 0.01 | 2.74 ± 0.03 | 3.28 ± 0.40 | $139^{+34}_{-29}$ | 1.80 ± 1.13 | $413^{+222}_{-94}$ |
| NHH6 | 55 | 0.99 | 84 | 2.52 | -57.5 | 60.02 | 2.57 ± 0.6 | 0.46 ± 0.01 | 8.93 ± 0.07 | 2.90 ± 0.42 | $166^{+34}_{-29}$ | 0.01 ± 1.63 | $12700^{+n/a}_{-12200}$ |
| NHH7 | 57.1 | 0.39 | 99 | -0.93 | -59.1 | 58.17 | 0.53 ± 0.02 | 0.50 ±0.01 | 15.6 ± 0.2 | 3.33 ± 0.57 | $135^{+44}_{-35}$ | 2.02 ± 1.38 | $390^{+257}_{-98}$ |
| NHH8 | 49.8 | 0.35 | 97 | 3.31 | -56.2 | 59.51 | 0.84 ± 0.03 | 0.39 ± 0.01 | 9.64 ± 0.10 | 3.64 ± 0.35 | $116^{+25}_{-20}$ | 5.14 ± 1.10 | $211^{+41}_{-31}$ |

1σ errors are given. Location is given as latitude and longitude in decimal degrees. The ³He/⁴He ratio, $R$, is shown relative to the ³He/⁴He ratio in air, $R_A$. *Adiabatic corrected. †Air corrected. ‡Corrected Δ18 from prototype Ultra).

**Extended Data Table 3 | Noble gas concentrations and ratios for the sampled wells (with 1σ errors), as well as measured carbon isotopic compositions of alkanes**

| Sample | $^{20}$Ne x$10^{-9}$ cm$^3$/cm$^3$ (STP) | $^{36}$Ar x$10^{-9}$ cm$^3$/cm$^3$ (STP) | $^{84}$Kr x$10^{-9}$ cm$^3$/cm$^3$ (STP) | $^{130}$Xe x$10^{-10}$ cm$^3$/cm$^3$ (STP) | $^{20}$Ne/$^{22}$Ne | $^{21}$Ne/$^{22}$Ne | $^{40}$Ar/$^{36}$Ar | $^{86}$Kr/$^{84}$Kr | $^{129}$Xe/$^{130}$Xe | $\delta^{13}$C-C$_2$ ‰VPDB | $\delta^{13}$C-C$_3$ ‰ VPDB | $\delta^{13}$C-nC$_4$ ‰ VPDB |
|---|---|---|---|---|---|---|---|---|---|---|---|---|
| O1 | 25.8 ± 0.5 | 84.9 ± 0.8 | 3.21 ± 0.09 | 3.46 ± 0.04 | 10.25 ± 0.03 | 0.0384 ± 0.0004 | 1910 ± 17 | 0.302 ± 0.002 | 6.59 ± 0.04 | | | |
| O2 | 26.9 ± 0.5 | 80.8 ± 0.8 | 3.66 ± 0.10 | 3.78 ± 0.05 | 10.18 ± 0.03 | 0.0371 ± 0.0003 | 1797 ± 16 | 0.302 ± 0.002 | 6.61 ± 0.04 | | | |
| O3 | 20.6 ±0.4 | 83.3 ± 0.8 | 1.81 ± 0.05 | 2.04 ± 0.03 | 10.42 ± 0.03 | 0.0405 ± 0.0004 | 2344 ± 20 | 0.302 ± 0.002 | 6.60 ± 0.04 | -27.3 | -17.74 | -22.54 |
| O4 | 10.5 ± 0.2 | 66.0 ± 0.6 | 3.20 ± 0.09 | 9.04 ± 0.10 | 10.19 ± 0.03 | 0.0385 ± 0.0003 | 1475 ± 13 | 0.302 ± 0.002 | 6.57 ± 0.04 | | | |
| O5 | 10.1 ±0.2 | 45.6 ± 0.4 | 2.07 ± 0.06 | 2.77 ± 0.03 | 10.42 ± 0.03 | 0.0409 ± 0.0004 | 2484 ± 22 | 0.302 ± 0.002 | 6.61 ± 0.04 | | | |
| O6 | 8.83 ± 0.17 | 46.0 ± 0.4 | 2.17 ± 0.05 | 2.83 ± 0.03 | 10.64 ± 0.03 | 0.0459 ± 0.0004 | 2766 ± 24 | 0.302 ± 0.002 | 6.59 ± 0.04 | -28.14 | -20.22 | -21.58 |
| NHH1 | | | | | | | | | | | | |
| NHH2 | 10.3 ± 0.2 | 44.5 ± 0.4 | 2.42 ± 0.06 | 2.12 ± 0.03 | 9.93 ± 0.03 | 0.0330 ± 0.0003 | 435.7 ±3.8 | 0.303 ± 0.002 | 6.51 ± 0.04 | -24.19 | -25.87 | -26.00 |
| NHH3 | 36.7 ± 0.7 | 176 ± 2 | 5.95 ± 0.16 | 7.39 ± 0.09 | 9.83 ± 0.03 | 0.0310 ± 0.0003 | 369.7 ± 3.2 | 0.302 ± 0.002 | 6.52 ± 0.04 | | | |
| NHH4 | 3.90 ± 0.07 | 66.0 ± 0.7 | 4.09 ± 0.11 | 6.04 ± 0.07 | 9.76 ± 0.03 | 0.0342 ± 0.0003 | 398.6 ± 3.5 | 0.302 ± 0.002 | 6.50 ± 0.04 | -35.37 | -26.52 | -27.58 |
| NHH5 | 3.49 ± 0.07 | 47.2 ± 0.4 | 5.23 ± 0.13 | 10.53 ± 0.11 | 9.87 ± 0.03 | 0.0353 ± 0.0003 | 573.1 ± 5.0 | 0.304 ± 0.002 | 6.47 ± 0.04 | -36.96 | -28.08 | -26.62 |
| NHH6 | 14.9 ± 0.3 | 79.1 ± 0.7 | 8.14 ± 0.22 | 6.15 ± 0.08 | 9.84 ± 0.03 | 0.0335 ± 0.0003 | 494.6 ± 4.3 | 0.304 ± 0.002 | 6.50 ± 0.04 | | | |
| NHH7 | 21.6 ± 0.4 | 112 ± 1 | 4.70 ± 0.12 | 3.99 ± 0.05 | 10.02 ± 0.03 | 0.0333 ± 0.0003 | 603.8 ± 5.3 | 0.304 ± 0.002 | 6.47 ± 0.04 | | | |
| NHH8 | 7.08 ± 0.13 | 78.6 ± 0.8 | 2.32 ± 0.06 | 4.81 ± 0.06 | 9.77 ± 0.03 | 0.0308 ± 0.0003 | 379.4 ± 3.3 | 0.303 ± 0.002 | 6.51 ± 0.04 | | | |
| Air | | | | | 9.81 | 0.0290 | 298.56 | 0.303 | 6.49 | | | |