## [Peer Review File · Nature]

Manuscript Title: Rapid microbial methanogenesis during CO₂ storage in hydrocarbon reservoirs

Reviewer Comments & Author Rebuttals

Reviewer Reports on the Initial Version:

Referee #1 (Remarks to the Author):

The manuscript "Rapid microbial methanogenesis during CO₂ storage in hydrocarbon reservoirs" explores to what extent CO₂ injected into an oil reservoir in the 1980's has been converted into methane by microbial methanogenesis. The authors conclude that 13-19% of the CO₂ was converted into methane over 40 years. To make their case, they present delta¹³C isotope compositions for methane and CO₂ in sampled gas, as well as CO₂/³He ratios. The latter show to what extent CO₂ has "disappeared" from the gas phase. Precipitation, dissolution and conversion are three possible explanations for CO₂ disappearance. They also measured the deuterium content of the methane. In combination, the isotope data show evidence for a higher proportion of biogenic methane and/or biological turnover of methane (via methanogenesis and the anaerobic oxidation of methane) in an oilfield that was subjected to EOR (enhanced oil recovery via CO₂ injection) – the Olla oilfield – than in a neighboring, geologically similar oilfield not subjected to EOR – the Nebo oilfield.

Conversion of a significant part of CO₂ to methane in CCS trials would be a big deal, both because of the higher warming potential and larger mobility of methane in the subsurface.

The research question is challenging to address, given the available data and samples. No historical samples are available I guess. The reservoirs are highly heterogeneous, as explained in the supplementary information. After CO₂ -EOR, Olla was also water-flooded. No information is shared on what type of water was used for flooding, Did it contain a lot of biodegradable material? Was it seawater, containing sulfate, that may have stimulated the anaerobic oxidation of methane (AOM)?

Previous work was done on samples from this oilfield, and previous work already provided evidence for methane and methanogenesis (a methanogenesis "hot spot"?) in this oil field. What I find missing is a clear explanation of what was done and concluded before and how the present work creates the leap forward from the previous understanding that I would expect for a paper in this journal. I would expect a paragraph near the start that delineates the present from the previous work. For example, how does the amplicon sequencing done relate to what was published before?

What I also found missing is a clear description of basic geochemistry. What was the actual methane concentration? If 13-19% of the CO₂ was bio-converted into methane, and the CO₂ concentration was 18% (v/v), the methane concentration should be very high. Strangely, it was not reported – at least I couldn't find it. According to the authors, most of the CO₂ was dissolved, yet the pH in the water was 7. This would require major leaching of cations from the sandstone matrix to compensate the production of bicarbonate. No data on brine composition is presented (apart from this bit buried on the supplement: "Notably, the Olla Field exhibits ideal environmental conditions for methanogenesis, in terms of pH (6.5-7.5), temperature (29.2-50.7 °C), salinity 306 (1094-1622 mM), sulfate (<0.6 mM/l) and iron concentrations (<0.6 mM/l expect for 1 site)."). This does not address my questions on CO₂ dissolution. I would expect a paragraph near the start explaining such simple observations before we dive into the much harder to interpret isotopic data.

With regard to the isotope data, I am not an expert, but I think I understand enough to agree that methane is in part produced biologically, and perhaps more so at Olla than at Nebo. However, I

think the data does not provide enough evidence to safely conclude that the CH₄ originated from the injected CO₂. Other interpretations are also possible. For example, CH₄ was produced from other components of the reservoir. This production might have been stimulated by CO₂ or water injection.

With regard to the microbiology data: Any reservoir contains methanogens, so no surprises there. Generally, natural activities are understood to be very low. So if a massive amount of CO₂ gets converted by methanogenesis, in 40 years, I would expect a major effect on the microbiome, enormous change, resulting in the microbial community to consist for at least 80% of methanogens. The data cannot show that, because different amplicon analyses were performed for Bacteria and Archaea. In the previous study, which combined both into a single analysis, this was not found. Actually, according to that study, CO₂ injection did not alter microbial communities significantly in the Olla oilfield. With regard to the postulated CH₄ cycling (methanogenesis + AOR) – the authors should identify which electron acceptor they believe could have sustained AOR activity. The authors also do not identify how much H₂ would be needed for the proposed methanogenesis and where this H₂ would come from, or what would have been its fate, if no CO₂ would have been injected. Essentially, if what the authors are proposing is true, the reservoir must have been CO₂ limited pre-EOR, with ample availability of hydrogen or hydrogen donors. It should be possible to find evidence for that in the Nebo oilfield.

Overall, with all the unknowns I think it might be very challenging to show conversion using data from this old CCS experiment that was never designed to answer the author's question. For example, if air-captured CO₂ would have been used, it would be easy to track conversion into methane by measuring its ¹⁴C content.

Overall, I found the manuscript challenging to read, with many minor imprecisions or awkward formulations. Often, I had to read sentences two times before I understood them.

Detailed comments:

L36: "promising" – please rephrase more neutrally, eg. use "possible" – There are many aspects to the technology that are not yet "promising".

L37 "CCS is effective..." seems like a sentence without information: Something is green is when something appears green?

L59 "... but evolves to progressively enriched ¹³C content with continued CO₂ consumption."

Formulation is misleading, as the relationship is not linear. The ¹³C content of the methane displays most of the increase only when the CO₂ is almost fully consumed. Rephrase or remove.

L69 "we assume" The authors should be explicit about what specifically was assumed about the "geochemical compositions". Some aspects must be known, for example, the fields are apparently in the same Wilcox Group formation (Figure 1).

L78 "involved with" => "associated with"

L82-85 rephrase for precision and readability. For example "enhanced mantle component"?

L92 and following: What do the authors mean with CO₂ trapping? Do they mean precipitation as carbonates? I see this is explained later on, needs restructuring for clarity.

L188-196 (16S metabarcoding/amplicon sequencing): (1) As Archaea and Bacteria were analyzed with different primersets, the data do not show enrichment of H₂/CO₂ methanogens relative to bacteria. (2) There are no convincing differences between samples from Olla and Nebo oilfields. One Nebo sample is the outlier, that sample may have a somewhat higher temperature, based on the taxa found. Reference 43 states, about the Olla oilfield, that "Microbial communities in CO₂-impacted and non-impacted samples did not significantly differ".

L197: "The integration of independent geochemical and microbiological constraints we provide here overwhelmingly supports our assertion that injected CO₂ is being consumed at Olla by microbial methanogenesis"

Sorry, but you did not convince me. I think the data does indeed support that methane is in part produced biologically, and perhaps more so at Olla than at Nebo. However, the data does directly show that injected CO₂ is getting converted into methane. With the data presented, methane could also be formed biologically from other sources. Because no historical data/samples are available, we need to be careful not to over-interpret differences between reservoirs, as significant within-reservoir heterogeneity also exists and may have contributed to observed differences.

I realize while reading that I'm missing basic information on the samples obtained/sampling procedure used. Surely, the authors also have brine samples to quantify bicarbonate dissolution? A few sentences about samples at the beginning of the results section would be good to add. Also, I would like to know from what depth the samples were obtained.

References should be reviewed one by one as I noticed in some cases more recent papers have been published that provide more up to date information, such as, about CO₂ mineralization:

Matter et al., (2016) Rapid carbon mineralization for permanent disposal of anthropogenic carbon dioxide emissions. *Science* 352, 1312-1314.

References 5 and 6, provided to this topic are 15 and 17 years old.

Referee #2 (Remarks to the Author):

General assessment

This manuscript presents a quite comprehensive set of data regarding the fate of CO₂ injected into an oil field for enhanced oil recovery (EOR). The data include noble gases, carbon isotopes of CO₂ and CH₄, clumped isotopes on CH₄, as well as gene sequencing of microbial communities. It is interpreted in terms of CO₂ removal by dissolution in groundwater and methanogenesis. The importance of the former process has been shown in a previous similar paper (ref. 4 in the manuscript), the occurrence of the latter process is the new finding presented here. Similar to the older paper, the noble gas data provide convincing evidence for a loss of CO₂, which here is accompanied by methanogenesis.

This importance of the whole topic comes from plans to store large quantities of anthropogenic CO₂ in geologic formations, i.e. Carbon Capture and Storage (CCS) or CO₂ sequestration in general. Although the study site used here is not a CCS site, the process of CO₂-EOR applied to one of the studied oil fields is a very good analogue. A strength of the present study is that the EOR-affected oil field can be compared to a nearby field without this disturbance, providing a good control case. The finding of significant amounts of methane formation in such a setting appears very relevant for CCS applications, as this process could be problematic for the safety of such operations, although such implications are hardly discussed in this manuscript. Given the importance of CO₂ sequestration as a possible future climate change mitigation strategy, the paper should be of interest for a wider audience and thus probably suitable for publication in *Nature*.

The paper is generally well written and referenced and makes a quite convincing case based on a very high quality set of data and methods. The conclusion appear valid, however, some details of the arguments and of their presentation have not become entirely clear to me, as pointed out in the remarks below. Overall, I think these are rather minor and technical issues, that the authors should be able to resolve. If so, I think the paper can be recommended for publication.

Remarks

1) My main difficulty with the manuscript is to understand exactly what the processes are that presumably have happened since CO₂ injection and how they change the carbon isotopes. This may be due to my lack of understanding of the situation, but I think that some precision in formulation would also help. It starts in the abstract: "We show that most of the injected CO₂ *remained dissolved* in the formation waters after the EOR project, but microbial methanogenesis converted as much as 13-19 % of the *remaining* fraction of CO₂ to CH₄." On lines 47-49 it is stated that about 30 % of the initially injected CO₂ *remained* in the reservoir. What exactly is meant with "remain" and "remaining fraction" in these sentences?

I try to formulate my understanding, which may be wrong: There is a total amount of injected gaseous CO₂, of which about 30 % have remained in the reservoir (not necessarily dissolved) after injection. A part of this "remaining fraction" has then been converted (or trapped) by dissolution and/or methanogenesis. According to Fig. 2a and statements on line 95, 39-89 % of this amount has been trapped, and the sub-fraction that went into methane is 13-19 %.

If that is correct, my formulation of the above sentence in the abstract would be something like: "We show that most of the injected CO₂ remaining in the oil field after the EOR project was dissolved in the formation waters, but microbial methanogenesis converted as much as 13-19 % of this CO₂ to CH₄." Remaining then refers to the 30 % of the initially injected CO₂ that is still around, part of which is dissolved and another part of the same transformed to methane.

2) The distinction of initial and remaining reservoirs become important in the methods section, where d₁₃C values of the CO₂ are modelled in equations (1) to (3). In my understanding, the above "remaining fraction" of CO₂ is now really the starting point, or a similar amount tied to sample O5. This is what I would now call the "initial" reservoir of (in my understanding gaseous) CO₂, from which the Rayleigh process starts. It has the isotopic composition indicated by d₁₃C_CO₂i. If correct, the parameter f in the model equations is not the trapped/converted fraction of the original CO₂ (lines 389 and 404), but rather the remaining (not trapped/converted) fraction of the original reservoir (modelled by sample O5).

3) I am uncertain about the definition of the fractionation factor α_m in Eqs. (2) and (4). As given on line 398, α_m is larger than 1. Then, $(\alpha_m - 1)$ and the entire last term of Eq. (2) is positive, and thus d₁₃C_CO₂ resulting from this process would be smaller than d₁₃C_CO₂i. However, the line for methanogenesis in Fig. 2a goes to higher d₁₃C_CO₂ values. I believe that the value for α_m in this equation should actually be the inverse of the given α_m , which seems to indicate the fractionation of CO₂ relative to CH₄. It depends on the direction of the conversion process you consider, and I think you need the fractionation of CH₄ relative to CO₂, which would be smaller than 1. This is also important for Eq. (4). With α_d larger than 1 and α_m smaller than 1, this equation models the transition from depletion by dissolution to enrichment by methanogenesis, as indicated in Fig. 2a.

4) There is definitely a minor mistake in Eq. (2), where it should read x1000 at the end, as in Eq. (3), not /1000. This is to transform from $(\alpha - 1)$ to epsilon in permille.

Minor corrections

Line 33: in th*is* type of ...

Line 412: Should this be Noble *Gas* Laboratory?

Lines 420, 423: μm and $^{\circ}\text{C}$.

Supplement

Line 27: provide clear evidence **that** the ...

Ext. Data Fig. 7: In the caption, M:D ratios of 0.055 and 0.60 are mentioned, in the figure, lines for 0.05 and 0.06 are indicated. Please clarify.

Line 280: these temperature*s* ...

Reviewer signature
Review by Werner Aeschbach

Referee #3 (Remarks to the Author):

Overall, this is a well written study summarizing the stable gas isotopes, gas composition, noble gases, and 16S rRNA gene sequencing data from the Olla and Nebo Hemphill fields. However, much of this work (stable gas isotopes, gas composition, and 16S rRNA gene sequencing) has been performed previously by McIntosh et al., 2010, Shelton et al., 2014, 2016 (frontiers in microbiology), 2016 (organic geochemistry). The noble gas applications are however novel to this oil field.

Some conclusions about the novelty of this study may be overinflated, and as discussed in the embedded comments, determination of where the samples in the Olla Oil Field came from (i.e., which producing strata) is vital in determining how the injected CO₂ impacted methanogenesis in this oil field. Those data and considerations of producing formations is currently lacking from the manuscript. These data are essential prior to publication of this work.

Referee #4 (Remarks to the Author):

I have reviewed with great interest the paper submitted to Nature by Tune et al. The paper is very well written and documented, and importantly it tackles a very important societal issue using a true multi-proxy approach. The field of carbon capture and storage (CCS) is in my opinion a key technology to mitigate anthropogenic greenhouse gases. The main challenges of CCS are finding secure storage locations, and monitoring for CO₂ escape. What this paper does is novel: it asks the question of the impact of the deep biosphere on the fate of CO₂. It does this by comparing two nearby oil fields, one produced with enhanced oil recovery (EOR, the Olla Oil Field) using CO₂ injection, and one (Nebo-Hemphill oil field) where CO₂ injection has never been used. The paper confirms that CO₂ has been trapped in the field produced with CO₂ EOR (i.e. residual trapping) but what is more impactful is that the authors also find that the proportion of CH₄ (methane) is higher in this field. By using stable isotopes, noble gases, organic clumped isotopes and genome sequencing, the authors demonstrate (in my opinion quite convincingly) that the CH₄ is of biogenic origin and is the result of the degradation of CO₂ by bacterial processes.

This has some tremendous implications: CH₄ is an even more potent (albeit short-lived) greenhouse gas in the atmosphere, and its flow properties in the subsurface (and thus the ability of a cap rock to retain it) are very different than that of CO₂. I completely agree with the authors that this has been overlooked, but needs to be taken into consideration for CO₂ injections in future!

I don't have major comments on the paper itself, it is very well written, the data and methods are well explained, and I cannot find flaws in any of the arguments (my expertise is in carbonate clumped isotopes, so perhaps an organic geochemist will have a different view here. But for me it sounds all very reasonable).

Perhaps the only thing I would have liked is for the authors to add one paragraph at the end that spells out the implications of CO₂ being transformed into CH₄ via biogenic pathways during CCS: it will not be clear to all readers of Nature what the challenge of this could be.

Author Rebuttals to Initial Comments:

We thank the reviews for their thoughtful and constructive reviews and are pleased that they are broadly supportive of this work being published. Direct responses to reviewer comments are outlined below in red text.

Reviewer #1

The manuscript “Rapid microbial methanogenesis during CO₂ storage in hydrocarbon reservoirs” explores to what extent CO₂ injected into an oil reservoir in the 1980’s has been converted into methane by microbial methanogenesis. The authors conclude that 13-19% of the CO₂ was converted into methane over 40 years. To make their case, they present delta¹³C isotope compositions for methane and CO₂ in sampled gas, as well as CO₂/³He ratios. The latter show to what extent CO₂ has “disappeared” from the gas phase. Precipitation, dissolution and conversion are three possible explanations for CO₂ disappearance. They also measured the deuterium content of the methane. In combination, the isotope data show evidence for a higher proportion of biogenic methane and/or biological turnover of methane (via methanogenesis and the anaerobic oxidation of methane) in an oilfield that was subjected to EOR (enhanced oil recovery via CO₂ injection) – the Olla oilfield – than in a neighboring, geologically similar oilfield not subjected to EOR – the Nebo oilfield. Conversion of a significant part of CO₂ to methane in CCS trials would be a big deal, both because of the higher warming potential and larger mobility of methane in the subsurface.

We thank the reviewer for this accurate summary of the paper, and their overall support of the study. We agree that this study has important broader climate implications and would indeed be important for CCS considerations.

The research question is challenging to address, given the available data and samples. No historical samples are available I guess. The reservoirs are highly heterogeneous, as explained in the supplementary information. After CO₂ -EOR, Olla was also water-flooded. No information is shared on what type of water was used for flooding, Did it contain a lot of biodegradable material? Was it seawater, containing sulfate, that may have stimulated the anaerobic oxidation of methane (AOM)?

The reviewer is correct that limited information is available on the historical injection, which we now clearly state in the supplementary material. However, we note that regardless of the source of the biological components, we present the case that the isotope systematics of CO₂ and ³He are unambiguous in showing microbial methanogenesis occurring within the field, which is the principal finding of our paper. Similarly, we show systematic differences between two fields that produce from the same reservoir intervals. First, Olla clearly contains high concentrations of post-injection CO₂ that has itself been heavily altered by microbial processes as indicated by the high δ¹³C-CO₂. Second, the methane clumped isotope signature of the Olla fluids is offset relative to the unperturbed Nebo Hemphill site. Given the differences in both CO₂ and CH₄ composition and isotopic signature, it is hard to reach any conclusion other than the injection and subsequent microbial consumption of CO₂ in Olla is responsible for these differences over very short, decadal

timescales. We do agree that there will be considerable future work needed, building on our findings, to identify the underlying mechanisms and rates driving this conversion.

Previous work was done on samples from this oilfield, and previous work already provided evidence for methane and methanogenesis (a methanogenesis “hot spot”?) in this oil field. What I find missing is a clear explanation of what was done and concluded before and how the present work creates the leap forward from the previous understanding that I would expect for a paper in this journal. I would expect a paragraph near the start that delineates the present from the previous work. For example, how does the amplicon sequencing done relate to what was published before?

Excellent point, we now provide a comprehensive summary of previous findings within the SOM. This is also referred to in the overview at the beginning of the main text.

What I also found missing is a clear description of basic geochemistry. What was the actual methane concentration? If 13-19% of the CO₂ was bio-converted into methane, and the CO₂ concentration was 18% (v/v), the methane concentration should be very high. Strangely, it was not reported – at least I couldn't find it.

Apologies for the oversight, the methane concentration is 57-89 mol %. This was in the SOM table 1 but has also been added to the main text.

According to the authors, most of the CO₂ was dissolved, yet the pH in the water was 7. This would require major leaching of cations from the sandstone matrix to compensate the production of bicarbonate. No data on brine composition is presented (apart from this bit buried on the supplement: “Notably, the Olla Field exhibits ideal environmental conditions for methanogenesis, in terms of pH (6.5-7.5), temperature (29.2-50.7 °C), salinity 306 (1094-1622 mM), sulfate (<0.6 mM/l) and iron concentrations (<0.6 mM/l expect for 1 site).”). This does not address my questions on CO₂ dissolution. I would expect a paragraph near the start explaining such simple observations before we dive into the much harder to interpret isotopic data.

This is a good point, we have added a sentence comparing the water chemistry observations at Olla and at Nebo-Hemphill and reference to a new supplementary table containing the formation water chemistry has been added. We have also added a dedicated section in the SOM about formation water chemistry. Chloride is the dominant anion (1122–1621 mmol/L) and sodium is the dominant cation (1108–1630 mmol/L), with concentrations typical of basinal brines (Kharaka and Hanor, 2003). Our formation water chemistry is consistent with that in Shelton et al., 2014. Our conclusion, that CO₂ dissolution is a significant sink is also reached by Shelton et al., 2014 (although we are now able to quantify this). The formation water pH response depends on cation availability, whether there is active fluid migration and the physical extent of the groundwater system. For example, in a fully open system that is part of a regional aquifer system we would not expect a substantial change in pH. The isotope systematics allow us to investigate the CO₂ behaviour without the added complexities of associated and multiple processes that are inherent with interpreting pH and cation availability or kinetics. Using the isotopic signals, we are able to unambiguously observe a significant amount of CO₂ loss, which make the case can only be explained by the combination of dissolution and methanogenesis.

With regard to the isotope data, I am not an expert, but I think I understand enough to agree that methane is in part produced biologically, and perhaps more so at Olla than at Nebo. However, I think the data does not provide enough evidence to safely conclude that the CH₄ originated from the injected CO₂. Other interpretations are also possible. For example, CH₄ was produced from other components of the reservoir. This production might have been stimulated by CO₂ or water injection.

It is unambiguous from the change in CO₂/³He that CO₂ has been lost from the gas phase. This correlates with CO₂ isotopic compositional change. We make the case in the text that, in addition, a wide range of other isotope systems (CO₂, CH₄, clumped and noble gas) within the Olla reservoir are also coherent - and all consistent with the combination of dissolution and microbial conversion of CO₂ to methane. If microbial methane was being formed only from hydrocarbon degradation, then the equilibrium observed between the CO₂ and CH₄ isotope systems and consistency across the noble gases and clumped isotopes would not be observed. Methane was certainly present before the CO₂ injection, and we do not argue that all methane is derived from microbial methanogenesis. As above, the high concentrations of CO₂ and highly positive δ¹³C-CO₂ requires that significant post-injection CO₂ remains in Olla, and that the residual CO₂ has been heavily altered by microbial and dissolution processing. Our work identifies and quantifies the respective roles of methanogenesis and dissolution, we agree with the reviewer that future work to understand what controls the rate of these processes, building on our paper, will be very important.

With regard to the microbiology data: Any reservoir contains methanogens, so no surprises there. Generally, natural activities are understood to be very low. So if a massive amount of CO₂ gets converted by methanogenesis, in 40 years, I would expect a major effect on the microbiome, enormous change, resulting in the microbial community to consist for at least 80% of methanogens. The data cannot show that, because different amplicon analyses were performed for Bacteria and Archaea. In the previous study, which combined both into a single analysis, this was not found. Actually, according to that study, CO₂ injection did not alter microbial communities significantly in the Olla oilfield.

The reviewer is correct that previous work has suggested that a community response is small/negligible on short timescales (e.g., Mu et al., 2014, Li et al., 2017), however these studies were notably conducted in quite different CO₂-injected environments (i.e., saline aquifers), so may not be analogous. For example, there are no studies, to our knowledge, systematically investigating the microbial community response over a decadal timescales and further studies, building on our findings, are certainly required in order to understand this. For example, there is no guarantee that changes to the microbial community composition would be retained over decadal timescales. Although a very different environment, temporal studies in the aftermath of the Deepwater Horizon Spill show a short term shift in the microbial communities, eventually returning to baseline values that existed before the event (E.g., Kessler et al 2011, Dubinsky et al., 2013, Joye and Kositha 2020). Indeed, Shelton et al (2018) speculate that in the Olla field '*...microbial communities in affected wells may have reverted back to pre-injection conditions over the ca. 40 years since the CO₂-EOR.*'

With regard to the postulated CH₄ cycling (methanogenesis + AOR) – the authors should identify which electron acceptor they believe could have sustained AOR activity. The authors also do not identify how much H₂ would be needed for the proposed methanogenesis and where this H₂ would come from, or what would have been its fate, if no CO₂ would have been injected. Essentially, if

what the authors are proposing is true, the reservoir must have been CO₂ limited pre-EOR, with ample availability of hydrogen or hydrogen donors. It should be possible to find evidence for that in the Nebo oilfield.

There was no detectable H₂ gas in the Olla field, compared to up to 0.69% at Nebo-Hemphill. However, H₂ is rarely observed in produced gases from hydrocarbon fields, likely as a result of its high reactivity. Despite this, several studies provide support for the abundance of available hydrogen in hydrocarbon reservoirs that could drive the microbial reactions we propose here. For example, it is argued that hydrocarbon C and H signature are obtained by “metastable cyclic equilibrium” through a network of radical chain reactions among the hydrocarbon gas species (Thiagarajan et al., 2020). This is predicted from theoretical models (e.g., Xiao et al., 2001), suggesting that hydrocarbon derived hydrogen is constantly exchanging within the system and would be available for CH₄ production (Thiagarajan et al., 2020).

Overall, with all the unknowns I think it might be very challenging to show conversion using data from this old CCS experiment that was never designed to answer the author’s question. For example, if air-captured CO₂ would have been used, it would be easy to track conversion into methane by measuring its ¹⁴C content.

We agree that dedicated multi-decadal studies looking into these processes would be great. However, given the immanency of CCS projects, we need answers to these sorts of questions now and therefore we need to make our interpretation using available analogues.

The notion of using radiocarbon in injected air to track conversion of injected CO₂ to CH₄ is an interesting idea. In our work we consider something broadly similar. We characterize the mantle-like signature in the injected CO₂ as a means to trace the injected CO₂. We detail in our paper how the change in the CO₂/³He ratio allows us to determine the amount of CO₂ loss and associated changes in δ¹³C allows us to quantify the amount of CO₂ consumed and converted to CH₄.

Overall, I found the manuscript challenging to read, with many minor imprecisions or awkward formulations. Often, I had to read sentences two times before I understood them.

We agree that several statements were overly confusing and have changed a number of sentences throughout the manuscript to improve clarity, following recommendations by this and the other reviewers.

Detailed comments:

L36: “promising” – please rephrase more neutrally, eg. use “possible” – There are many aspects to the technology that are not yet “promising”.

Changed

L37 “CCS is effective...” seems like a sentence without information: Something is green is when something appears green?

This sentence has been removed

L59 "... but evolves to progressively enriched ^{13}C content with continued CO_2 consumption."
Formulation is misleading, as the relationship is not linear. The ^{13}C content of the methane displays most of the increase only when the CO_2 is almost fully consumed. Rephrase or remove.

Wording has been changed to "... but evolves to an enriched $\delta^{13}\text{C}$ content with continued CO_2 consumption."

L69 "we assume" The authors should be explicit about what specifically was assumed about the "geochemical compositions". Some aspects must be known, for example, the fields are apparently in the same Wilcox Group formation (Figure 1).

We make this assumption based upon the similarities in the molecular geochemistry of the oil, both fields producing from within the middle Wilcox Group. More information about geochemical assumptions are now clearly stated in the supplementary information (SI2) which is referred to in this sentence ("Given the close proximity and comparable geologic history (including hydrocarbon production from similar reservoirs, SI.1, SI.2), we assume that Nebo-Hemphill and Olla originally had comparable geochemical compositions"). However this assumption does not impact our findings of rapid methanogenesis of the injected CO_2 within the Olla field, as we use the most pristine sample from Olla to calculate this, rather than previously published data.

L78 "involved with" => "associated with"

This has been changed.

L82-85 rephrase for precision and readability. For example "enhanced mantle component"?

Changed to 'Enhanced mantle-derived noble gases contribution'

L92 and following: What do the authors mean with CO_2 trapping? Do they mean precipitation as carbonates? I see this is explained later on, needs restructuring for clarity.

Initial mention of trapping around $\text{CO}_2/{}^3\text{He}$ has been changed to 'removal'.

L188-196 (16S metabarcoding/amplicon sequencing): (1) As Archaea and Bacteria were analyzed with different primersets, the data do not show enrichment of H_2/CO_2 methanogens relative to bacteria. (2) There are no convincing differences between samples from Olla and Nebo oilfields. One Nebo sample is the outlier, that sample may have a somewhat higher temperature, based on the taxa found. Reference 43 states, about the Olla oilfield, that "Microbial communities in CO_2 -impacted and non-impacted samples did not significantly differ".

Please see our earlier reply to microbial community dynamics. Without a dedicated decadal-scale study of the microbiology we are unable to say with certainty whether there has been a shift in the microbial communities within the Olla field. However, our data does show that in this complex microbial system the right microbial communities are present for the processes that are occurring and this is consistent with the isotopic evidence we present that CO_2 is being consumed. Reference 43 makes their conclusions based upon the assumption that the injected CO_2 remained within the 2800ft sst producing formation and therefore compared different producing zones within the Olla

field to reach their conclusions. From the CO₂ concentration data and additional noble gas data presented here, we show that injected CO₂ is in fact present in all sampled producing zones.

L197: “The integration of independent geochemical and microbiological constraints we provide here overwhelmingly supports our assertion that injected CO₂ is being consumed at Olla by microbial methanogenesis”

Sorry, but you did not convince me. I think the data does indeed support that methane is in part produced biologically, and perhaps more so at Olla than at Nebo. However, the data does not directly show that injected CO₂ is getting converted into methane. With the data presented, methane could also be formed biologically from other sources. Because no historical data/samples are available, we need to be careful not to over-interpret differences between reservoirs, as significant within-reservoir heterogeneity also exists and may have contributed to observed differences.

We are not sure how to respond to this comment. The reviewer states above ‘*However, the data does directly show that injected CO₂ is getting converted into methane.*’ - which is the assertion we have made.

As stated above, methane was present within the system prior to CO₂ injection. The focus of our paper is not on the original source of the methane, but the way in which its isotopic systems were subsequently processed. We agree that some of the methane could have formed via other microbial pathways, such as from hydrocarbon biodegradation. However, we show a systematic difference in the CO₂ content in a field that has been subjected to CO₂ flooding. The initial injection report (Moffet, 1990) indicates that CO₂ concentrations greater than 10% at Olla are indicative of injected CO₂, and indeed, concentrations >10% are present in all the producing zones. We also show that this CO₂ has clear evidence of heavy alteration via microbial processes as required by such positive $\delta^{13}\text{C}$ -CO₂ values, which in turn require microbial cycling and net consumption of CO₂ within the Olla field.

I realize while reading that I’m missing basic information on the samples obtained/sampling procedure used. Surely, the authors also have brine samples to quantify bicarbonate dissolution? A few sentences about samples at the beginning of the results section would be good to add. Also, I would like to know from what depth the samples were obtained.

With thanks for pointing this out. Depths alongside producing formations have been added to supplementary table 3. Sampling procedures can be found within the methods. We have also added a section in the supplementary information regarding the chemistry of the formation waters. The formation water from Olla is part of a regional aquifer system and is acting as an open system, as a result we are unable to quantify the amount of bicarbonate dissolution.

References should be reviewed one by one as I noticed in some cases more recent papers have been published that provide more up to date information, such as, about CO₂ mineralization:

Matter et al., (2016) Rapid carbon mineralization for permanent disposal of anthropogenic carbon dioxide emissions. *Science* 352, 1312-1314.

References 5 and 6, provided to this topic are 15 and 17 years old.

We have added the Matter reference and have reviewed the remaining references to ensure they provide up-to-date information.

Reviewer #2

General assessment

This manuscript presents a quite comprehensive set of data regarding the fate of CO₂ injected into an oil field for enhanced oil recovery (EOR). The data include noble gases, carbon isotopes of CO₂ and CH₄, clumped isotopes on CH₄, as well as gene sequencing of microbial communities. It is interpreted in terms of CO₂ removal by dissolution in groundwater and methanogenesis. The importance of the former process has been shown in a previous similar paper (ref. 4 in the manuscript), the occurrence of the latter process is the new finding presented here. Similar to the older paper, the noble gas data provide convincing evidence for a loss of CO₂, which here is accompanied by methanogenesis.

This importance of the whole topic comes from plans to store large quantities of anthropogenic CO₂ in geologic formations, i.e. Carbon Capture and Storage (CCS) or CO₂ sequestration in general. Although the study site used here is not a CCS site, the process of CO₂-EOR applied to one of the studied oil fields is a very good analogue. A strength of the present study is that the EOR-affected oil field can be compared to a nearby field without this disturbance, providing a good control case. The finding of significant amounts of methane formation in such a setting appears very relevant for CCS applications, as this process could be problematic for the safety of such operations, although such implications are hardly discussed in this manuscript. Given the importance of CO₂ sequestration as a possible future climate change mitigation strategy, the paper should be of interest for a wider audience and thus probably suitable for publication in Nature.

The paper is generally well written and referenced and makes a quite convincing case based on a very high quality set of data and methods. The conclusion appear valid, however, some details of the arguments and of their presentation have not become entirely clear to me, as pointed out in the remarks below. Overall, I think these are rather minor and technical issues, that the authors should be able to resolve. If so, I think the paper can be recommended for publication.

We thank the reviewer for their summary and are happy that they found this work convincing and of interest to a wide audience.

Remarks

1) My main difficulty with the manuscript is to understand exactly what the processes are that presumably have happened since CO₂ injection and how they change the carbon isotopes. This may be due to my lack of understanding of the situation, but I think that some precision in formulation would also help. It starts in the abstract: "We show that most of the injected CO₂ *remained dissolved* in the formation waters after the EOR project, but microbial methanogenesis converted as much as 13-19 % of the *remaining* fraction of CO₂ to CH₄." On lines 47-49 it is stated that about 30 % of the initially injected CO₂ *remained* in the reservoir. What exactly is meant with "remain" and "remaining fraction" in these sentences?

Thank you for bringing this lapse in clarity to our attention, within this sentence we are referring to injected CO₂ that remained after the CO₂ injection project ceased, and then the fraction of this amount that was consumed by net microbial methanogenesis. We have amended this sentence to 'We show that microbial methanogenesis converted as much as 13-19 % of the injected CO₂ to CH₄ and up to an additional 74% of CO₂ was dissolved.' to remove any uncertainty.

I try to formulate my understanding, which may be wrong: There is a total amount of injected gaseous CO₂, of which about 30 % have remained in the reservoir (not necessarily dissolved) after injection. A part of this "remaining fraction" has then been converted (or trapped) by dissolution and/or methanogenesis. According to Fig. 2a and statements on line 95, 39-89 % of this amount has been trapped, and the sub-fraction that went into methane is 13-19 %.

This is correct.

If that is correct, my formulation of the above sentence in the abstract would be something like: "We show that most of the injected CO₂ remaining in the oil field after the EOR project was dissolved in the formation waters, but microbial methanogenesis converted as much as 13-19 % of this CO₂ to CH₄." Remaining then refers to the 30 % of the initially injected CO₂ that is still around, part of which is dissolved and another part of the same transformed to methane.

The reviewer is correct in the formulation and we have amended this sentence as stated above.

2) The distinction of initial and remaining reservoirs become important in the methods section, where d13C values of the CO₂ are modelled in equations (1) to (3). In my understanding, the above "remaining fraction" of CO₂ is now really the starting point, or a similar amount tied to sample O5.

As we do not have the exact concentration and d13C of CO₂ remaining, we are using our most pristine sample (O5) as our starting composition.

This is what I would now call the "initial" reservoir of (in my understanding gaseous) CO₂, from which the Rayleigh process starts.

Yes this is what we are referring to as initial in the models.

It has the isotopic composition indicated by d13C_CO2i. If correct, the parameter f in the model equations is not the trapped/converted fraction of the original CO₂ (lines 389 and 404), but rather the remaining (not trapped/converted) fraction of the original reservoir (modelled by sample O5).

The reviewer is correct and f as the remaining fraction is what had actually been used within the models. Therefore, the text has been corrected, but no correction was required to the models and our results remain unchanged.

3) I am uncertain about the definition of the fractionation factor α_m in Eqs. (2) and (4). As given on line 398, α_m is larger than 1. Then, $(\alpha_m - 1)$ and the entire last term of Eq. (2) is positive, and thus d13C_CO₂ resulting from this process would be smaller than d13C_CO2i. However, the line for methanogenesis in Fig. 2a goes to higher d13C_CO₂ values. I believe that the value for α_m in this equation should actually be the inverse of the given α_m , which seems

to indicate the fractionation of CO₂ relative to CH₄. It depends on the direction of the conversion process you consider, and I think you need the fractionation of CH₄ relative to CO₂, which would be smaller than 1. This is also important for Eq. (4). With α_d larger than 1 and α_m smaller than 1, this equation models the transition from depletion by dissolution to enrichment by methanogenesis, as indicated in Fig. 2a.

We thank the reviewer for pointing this typo out. α_m has been modified to the correct value and now reflects what has been used within the models. The correction of this typo has no effect on the results of the study.

4) There is definitely a minor mistake in Eq. (2), where it should read x1000 at the end, as in Eq. (3), not /1000. This is to transform from ($\alpha - 1$) to epsilon in permille.

Equation 2 has been changed. The correction of this typo has no effect on the results of the study. We thank the reviewer for pointing this out.

Minor corrections

Line 33: in th*is* type of ...

This has been corrected to 'these types'

Line 412: Should this be Noble *Gas* Laboratory?

The official laboratory name is the "Noble Laboratory".

Lines 420, 423: μm and $^{\circ}\text{C}$.

These have been changed

Supplement

Line 27: provide clear evidence *that* the ...

Added

Ext. Data Fig. 7: In the caption, M:D ratios of 0.055 and 0.60 are mentioned, in the figure, lines for 0.05 and 0.06 are indicated. Please clarify.

The values in the caption and on the figure are now consistent.

Line 280: these temperature*s* ...

Added

Reviewer signature

Review by Werner Aeschbach

We thank Prof Aeschbach for his comprehensive and supportive review, which, has greatly improved the manuscript.

Referee #3 (Remarks to the Author):

Overall, this is a well written study summarizing the stable gas isotopes, gas composition, noble gases, and 16S rRNA gene sequencing data from the Olla and Nebo Hemphill fields. However, much of this work (stable gas isotopes, gas composition, and 16S rRNA gene sequencing) has been performed previously by McIntosh et al., 2010, Shelton et al., 2014, 2016 (frontiers in microbiology), 2016 (organic geochemistry). The noble gas applications are however novel to this oil field.

Some conclusions about the novelty of this study may be overinflated, and as discussed in the embedded comments, determination of where the samples in the Olla Oil Field came from (i.e., which producing strata) is vital in determining how the injected CO₂ impacted methanogenesis in this oil field. Those data and considerations of producing formations is currently lacking from the manuscript. These data are essential prior to publication of this work.

We thank the reviewer for their constructive and careful review of our manuscript. We agree that a great deal of excellent work has previously been done on this system, particularly with respect to CO₂ isotopes – which we have made great effort to fully acknowledge. Our contribution provides a new level of understanding and utility by extending the available data to include noble gas and clumped isotope constraints to these existing studies. Shelton and McIntosh found elevated $\delta^{13}\text{C-CO}_2$ and $\delta^{13}\text{C-DIC}$ and suggested microbial methanogenesis was occurring within the field. They suggested that the ideal environmental conditions (i.e., pH, salinity, substrate availability) caused greater rates of methanogenesis within the Olla field compared to other fields in the area. We build on this work to show that the injected CO₂ is being consumed and calculate the rate of this conversion. This unique dataset allows us to link CO₂ injection directly to methane production. In this way we are able to quantify the amount of CO₂ being converted to methane (by methanogenesis) and the rate at which it is occurring, for the first time in a natural system. We think this has important implications for future CCS studies and hope this short explanation and direct responses to the embedded comments are enough to convince the reviewer that our results are a significant finding.

The producing strata from which our samples are derived has been added to SOM table 1. Our new noble gas data allow us to demonstrate communication between the producing strata from the injected CO₂-derived mantle noble gas signature in all Olla samples from all lithologies sampled, as well as in quantifying the CO₂ consumption, we have added a paragraph to the SOM information to help improve the clarity surrounding the presence of CO₂.

We see a significant difference between the noble gas signatures (helium isotopes: $t_{\text{test}(11)} = 28.16$, $p\text{-value} = 1.3 \times 10^{-11}$) at Olla and Nebo Hemphill, with the Olla field displaying a much higher mantle fluid contribution. We also find that samples with the highest CO₂ concentrations have CO₂/³He ratios within the mantle range. There is no structure in the region that could explain the enhanced mantle component in the Olla field, but not the Nebo field, as the only significant difference between the Olla and Nebo Hemphill is the presence of injected CO₂. Noble gas data from close to the Black Lake Oil Field has shown the Sabine Island Uplift (for which the Black lake Oil Field lies on

the edge) is a conduit to mantle fluids in the region (Byrne et al., 2020). We agree that noble gas data from the black lake field would further strengthen our argument, however using the noble gas and $\text{CO}_2/{}^3\text{He}$ ratio from this study and previous studies near the Sabine Island Uplift, is fully consistent with the injected fluids having a mantle signature. This is nevertheless, largely irrelevant to the main results of our study, which quantify the change in geochemical composition relative to an in-situ 'most pristine' (O5) sample howsoever originally derived - and identify the process of change.

We find that noble gas isotope ratios and abundance (including those which are mantle derived) are consistent between the producing strata, suggesting fluid communication between the zones. Noble gases can indicate vertical fluid migration between formations, as has been recently shown in Byrne et al., 2018. This communication is reflected in other geochemical parameters; for example, Olla and Nebo have similar molecular geochemistry yet Olla has high CO_2 concentrations. The initial CO_2 injection report by Hunt petroleum (Moffet, 1990) states that: 'mole percentages in excess of 10% more than likely indicate production of injected CO_2 '. Only 1 sample outside of 2800ft sandstone has CO_2 concentrations less than 10% (i.e., all other CO_2 concentrations in this zone are >10%). Notably, concentrations >10% in other producing zones were also reported by Shelton et al., 2014. These signatures are consistent with injected CO_2 migration across the different reservoirs and remaining within the formations upon sampling.

Comments from text:

Line 40: proportion → portion

Change accepted

Line 44: Add USA

Done

Line 45: Compare what? Right now, it appears you are comparing the behaviour of CO_2 in both fields, which may or may not be correct.

We are comparing the (bio)geochemical characteristics (noble gas, stable, clumped and microbial analysis) of the 2 sites in order to investigate the behaviour of CO_2 within the fields. We have modified the sentence to read '... and compare its (bio)geochemical composition with the adjacent Nebo-Hemphill Oil Field...'. We think this improves the clarity of the sentence.

Line 46: Why is Olla Oil Field capitalized but Nebo-Hemphill oil field isn't? Please be consistent.

Capitalisation has been changed to be consistent throughout

Close proximity geographically? Stratigraphically?

In this sentence it was meant geographically and has been clarified within the text. However, it is also important to note that these units also produce from the same reservoirs in the middle Wilcox group and have broadly similar geological histories.

Line 47-48: What is 'the Olla reservoir?'" The Olla produces from many stratigraphic units, you need to be clear about this.

Changed to reservoirs to reflect multiple producing formations within the field.

Change 'is estimated to have remained' to 'has been retained'

Done

Line 50: Same issue here with capitalisation

Capitalisation has been changed to be consistent throughout

Line 58: This work has previously been done by McIntosh and Shelton. I rearranged the text to be clear that your contribution is unique in that you add noble gas and clumped isotope data to this study area.

Change accepted

Line 58: Or change to: given it sequesters CO₂ via producing CH₄ and H₂O

We have not added this sentence as due to word limit constraints.

Line 66: Check capitalisation

Capitalisation has been changed to be consistent throughout

Line 71: add 'of formation fluids'

You have yet to state exactly which producing strata were injected with CO₂ in the Olla oil field. This is critical to assess how the CO₂ was sequestered.

Producing formations have now been added to Supplementary Table 1. However, as stated above, we see that injected CO₂ is present within all the sampled producing zones.

Line 72: 'Is this correct? 80s to 20s is closer to 40 years.'

There were 29 years between the CO₂ injection ceasing (1986) and sample collection (2015).

'Again, are CO₂ concentrations high in ALL Olla Field producing strata, or just the sand that was injected with CO₂. This is critical.'

CO₂ concentrations are high across all the producing strata at Olla. As stated above, we find that noble gas isotope ratios and abundance (including those which are mantle derived) are consistent between the producing strata, suggesting fluid communication between the zones. Noble gases can

indicate vertical fluid migration between formations, as has been recently shown in Byrne et al., 2018. This communication is reflected in other geochemical parameters; for example, Olla and Nebo have similar molecular geochemistry yet Olla has high CO₂ concentrations. The initial CO₂ injection report by Hunt petroleum (Moffet, 1990) states that: 'mole percentages in excess of 10% more than likely indicate production of injected CO₂'. Only 1 sample outside of 2800ft sandstone has CO₂ concentrations less than 10% (i.e., all other CO₂ concentrations in this zone are >10%). Notably, concentrations >10% in other producing zones were also reported by Shelton et al., 2014. These signatures are consistent with injected CO₂ migration across the different reservoirs and remaining within the formations upon sampling.

Line 75: 'How did you get these data? Did you sample the Black Lake Field? Are you assuming that the isotopic composition of the produced gas at Black Lake Field has remained the same over the past 40 years? Is that a reasonable assumption?'

We assume that the isotopic composition of the CO₂ at Black Lake has remained the same between injection and sampling which is a common assumption in previous publications. Notably, the same assumption was made for Black Lake by Shelton et al., 2014, which is where we have taken δ¹³C-CO₂ measurements from (this has now been referenced). We also note that even if this assumption is invalid, significant microbial methanogenesis would be required for the isotopic composition at Olla to become so enriched, which we see no ancillary evidence for, so deem unlikely.

Line 76: 'This would also be exceptionally true if you sampled strata other than the injected strata. This has to be clarified.'

The sampled strata have been added to SOM Table 1. From the noble gas and CO₂ concentration data in this study, we conclude that injected CO₂ is present within all the producing strata sampled which is in contrast to the previous work on the field.

Line 82: sites → fields

Changed

Line 86: Add field

Added

Line 91-94: "While two samples at Olla (O5 and O6) have CO₂/³He within the mantle range (2±1x10⁹), consistent with injection of mantle derived fluids into the Olla Field, the majority of the ratios are lower and indicate CO₂ trapping/methanogenesis (Fig.2a, Supplementary Table 1)."

What strata are O5 and O6 from compared to other strata? Were O5 and O6 collected from the 2800' sand? If not, this would be an easy explanation, as those strata may not have 'observed' the injected CO₂.

O5 was collected from the 2800ft sandstone while O6 is from H2-HD, producing zones have been added to SOM table 1 for reference. Although sample O5 has the highest CO₂/³He and is from the

2800ft sst, the other sample from this formation has the 2nd lowest CO₂/³He (2.47x10⁸). We expect that all producing strata have injected CO₂ present (see general statement above).

Line 101: Add space

Done

Line 105: If you use the word significantly, you need some sort of p value to prove that the difference is indeed significant. I do not see any statistical evidence for the use of this word.

We are comparing that range to what would be expected from dissolution. Dissolution lowers the δ¹³C-CO₂, however as the exact numbers are unknown, no statistical test could have been completed. Therefore, we have removed the “significantly” from this sentence.

Line 107 (Figure 2): Look at capitalization of Olla Field here. Can you change “injected material” to “injected gas”? Injected material makes it sound like lots of things were injected downhole.

Figure 2 has been updated as per suggestions. Figure 3 and extended figure capitalisation has also been changed to be consistent.

Line 108: Again, this isn't true. The entire field wasn't flooded with CO₂.

As stated above, we find that noble gas isotope ratios and abundance (including those which are mantle derived) are consistent between the producing strata, suggesting fluid communication between the zones. This communication is reflected in other geochemical parameters; for example, Olla and Nebo have similar molecular geochemistry yet Olla has high CO₂ concentrations. The initial CO₂ injection report by Hunt petroleum (Moffet, 1990) states that: ‘mole percentages in excess of 10% more than likely indicate production of injected CO₂’. Only 1 sample outside of 2800ft sandstone has CO₂ concentrations less than 10% (i.e., all other CO₂ concentrations in this zone are >10%). Notably, concentrations >10% in other producing zones were also reported by Shelton et al., 2014. These signatures are consistent with injected CO₂ migration across the different reservoirs and remaining within the formations upon sampling.

We have modified the text to make it more clear that evidence for injection is observed throughout all reservoirs, rather than just within the 2800ft sst.

Line 110: “Why? Because it is the most pristine sample? What other evidence do you have for this one being the “most pristine”?”

The reviewer is correct. The tick marks for CO₂ loss are given relative to sample O5 as this is the most pristine sample. We arrive at the conclusion of O5 being the most pristine sample as it has the highest CO₂/³He ratio (following methodology adopted by Gillfillan et al., 2009). CO₂/³He ratios in the mantle are typically 2±1 x10⁹, while crustal CO₂ has a higher ratio. Any decrease in the CO₂/³He ratio therefore represents CO₂ loss from the system (e.g., Sherwood Lollar et al., 1997; Gillfillan et al., 2009). So regardless of the CO₂ loss mechanisms within Olla, O5 represents the sample with the least

CO₂ loss and thus the most pristine sample. In addition, it has the lowest $\delta^{13}\text{C-CO}_2$, again suggesting it is the least modified of all the samples.

Line 113: change 'showing dissolution to account for' to 'showing that dissolution accounts for'

Change accepted.

Line 116: Again, this is not true. ONLY the 2800' sand was flooded with CO₂, not the entire producing unit. If you sampled strata older than the 2800' sand and make the assumption that the CO₂ did not travel to strata beneath it, and you are seeing the same trends in older strata as you are in 2800' sand samples, how can you justify that the injected CO₂ did anything to the system? This is a critical point that is absolutely lacking from this manuscript.

We appreciate the position of the reviewer in understanding the distribution of CO₂ in the Olla field and understand why they could be confused based on the results of previous studies here.

However, we clearly show high concentrations of CO₂ in all producing intervals at Olla that we sampled. As stated above, the initial CO₂ injection report by Hunt petroleum (Moffet, 1990) states that: 'mole percentages in excess of 10% more than likely indicate production of injected CO₂'. Only 1 sample outside of 2800ft sandstone has CO₂ concentrations less than 10% (i.e., all other CO₂ concentrations in this zone are >10%). Notably, concentrations >10% in other producing zones were also reported by Shelton et al., 2014. These signatures are consistent with injected CO₂ migration across the different reservoirs and remaining within the formations upon sampling – a conclusion augmented by the presence of mantle-derived noble gases in these formations. If such high concentrations of CO₂ were present in this field prior to injection, CO₂ would not have been extracted and transported from the Black Lake field and would have instead have been extracted from the producing fluids of CO₂ rich fluids at Olla. This of course was not the case, and we are therefore confident that previous studies have mistakenly reported an absence of injected CO₂ in intervals other than the originally intended 2800" sandstone reservoir. Noble gas compositions within the Olla field are also consistent between the producing strata, suggesting fluid communication between the zones.

Line 121-123: Citation for this statement?

Reference to supplementary information has been made

Line 135-136: "I disagree. Fluids produced from the 2800' sand (the sand injected with CO₂ for ERO) would likely geochemically resemble strata within the Olla Field that were not injected with CO₂ much more so than those from Nebo Hemphill. Unless you only sampled the 2800' sand in Nebo-Hemphill, in which that may be another suitable analogue."

As with previous comments focussed on this concern, our findings suggest that all strata within the Olla field have been affected by injection.

Line 138: Get rid of underline

Removed

Line 140: offset- What does this mean? Do you mean a difference?

Offset has been changed to difference

Line 144: I do not understand this. It appears the Olla is much colder than expected for thermodynamic isotopic equilibrium. $29.2 \ll 44.6$???

This point is well taken, but there is very rarely perfect agreement in natural systems. Notably, we do not claim there is complete thermodynamic equilibrium, but instead note that there is an overlap between the clumped temperatures (44.6-76.3) and reservoir temperatures (29.2-50.7 °C). We were pleased to see overlap, as it provided additional validation for our modelling approach.

Line 148: Where is your evidence that this is occurring?

The sentence is 'The carbon isotopes within the system appear to be approaching equilibrium under reservoir conditions as a result of microbial cycling of carbon (i.e., methanogenesis and anaerobic oxidation of methane (AOM)).'. We are suggesting AOM here more as we need a mechanism of cycling the carbon, which is consistent with the clumped isotope systematics discussed later on. We have made reference at this point to the clumped isotopes presenting the evidence for this.

Line 153: add respectively

Added

Line 160: For the Olla oil field, we expect the reservoir temperature during thermogenic [A1] methane generation to have exceeded $163 \pm 18^\circ\text{C}$ from independent maturity constraints provided by biomarkers⁴¹

[A1]Is this correct? Otherwise, I do not understand this statement.

We have modified the sentence to include temperatures, so it makes more sense.

Line 168: OK, I see some evidence now. Can something be presented up front to justify the previous suggestion on AOM?

Reference has been made at the first mention of AOM to the evidence for the process in the clumped isotopologues.

Line 181: I do not understand the difference between the light grey arrows and the dark grey arrow. Can these be explained better in the figure caption, please?

We have modified the caption to 'Arrows represent the theoretical trends for methanogenesis (dark grey) and anaerobic oxidation of methane (AOM, light grey)³⁴.' to improve clarity regarding the arrows.

Line 183: Add 'produced'

added

Line 184: change heavy d13C of propane to positive d13C propane values

We have changed heavy to 'elevated'

Line 186: Add d13C

Added

Line 189: Add 'produced water from'

Added

Line 194: Change running to performing

Change accepted

Line 198-200: I do not necessarily disagree with this statement, but as I stated before, it is critical to mention which reservoirs in the Olla Field you sampled. For example, if you observed these same trends in reservoirs older than the 2800' sand, then it would be difficult to conclude that injected CO₂ was being consumed, as there would theoretically be no injected CO₂ in any producing unit other than the one that was flooded with CO₂, the 2800' sand.

As with previous comments focussed on this concern, our findings suggest that all strata within the Olla field have been affected by injection.

Line 200-202: This also isn't necessarily true. The Shelton et al., 2014 and 2016 papers did state that conversion of injected CO₂ to CH₄ was occurring in the 2800' sand of the Olla Oil Field, but that dissolution of CO₂ into formation water was the dominant trapping mechanism, which you also find here. This statement is unfortunately too bold.

We have built upon the work by Shelton and McIntosh identifying both methanogenesis and dissolution within the field. With new data across multiple isotope systems we quantify the amount and constrain the rate at which methanogenesis is occurring in response to injection.

Line 211: Add values

added

Line 224-225: Why do you think that is?

The difference between the rates calculated within this study and those of CO₂ reduction following methanogenic oil degradation by hydrogenotrophic methanogens from lab cultures could have multiple causes. These include:

- Firstly, the lab cultured study investigates CO₂ reduction during methanogenic oil degradation, rather than directly from CO₂. Having CO₂ readily available rather than oil degradation to begin with will likely result in differences. We are unaware of any studies of the rate of microbial methanogenesis directly from CO₂ in lab cultures under petroleum reservoir conditions, which would have served as a better comparison. This is likely a very different process. Therefore, differences between natural and lab rates of microbial processes are a common finding albeit with different microbes and environments (e.g., Weissman et al., 2021, Palkova 2004).
- Secondly, we chose to compare our rates to that of Gray et al 2009 as their lab conditions (i.e., T, S) are most like that of within the Olla reservoir. However, predicted rates of methanogenesis from lab experiments vary by 12 orders of magnitude (e.g., Lee et al., 2012, Yoshioka et al., 2009, Brook Avery et al., 2003, Gray et al 2009, Adler et al., 2011, Jones and Paynter 1980). It is therefore not surprising we do not find a rate that matches laboratory simulations.

Line 230: When was this discussed in your text? I do not see any mention of d13C-DIC in your manuscript. Also, can you provide some context for this field? Is it also located in Louisiana? What are the producing strata?

This field is in California (see line 228). CO₂ injection was into the Belridge Diatomites and the producing formation has been added to the sentence.

Line 380: This is assuming that all of your samples were collected from the same producing unit in the Olla Field. If not, you need to reconsider which samples are actually pristine re: CO₂ injection.

As with previous comments focussed on this concern, our findings suggest that all strata within the Olla field have been affected by injection.

Line 383: What is dissolution methanogenesis?

Changed to dissolution or methanogenesis

Line 396: change 'is considered to be 1.068+/- 0.003, consistent with..' to 'is 1.068+/-0.003 which is consistent with..'

Done

Line 399: Change 'Within the olla system, it is likely there is' to 'Within the Olla field, it is likely that'

Done

Line 400: This needs to be cited.

This sentence has now been removed from the text

Line 410: add gas

added

Line 411: Add 'directly from the wellhead'

added

Line 423 *S*terivex

Changed

Line 431: *2x* 250 bp paired end sequencing'

This has been changed.

Line 436: *N*aive Bayes taxonomy

Changed

Supplementary Information:

Line 8: Again, please do not use this word unless you have the statistical analysis to back it up.

We find that for the helium isotopic composition (for which mantle contributions are most easily identified) there is a significant difference between fields ($t_{test(11)} = 28.16$, $p\text{-value} = 1.3 \times 10^{-11}$), this has been added to the text.

Line 27-28: isotope*o*gues.

Corrected

Add 'clear evidence that produced methane from the olla field is microbial in origin'

This has been added.

Line 46: I am concerned why you are plotting every sample from reference 5. Some of those samples were collected in strata not impacted by the CO₂ injection. Why are you comparing them to samples collected in CO₂ impacted strata and only making conclusions about injected CO₂ converted to methane? There was no injected CO₂ in many of those samples.

As with previous comments focussed on this concern, our findings suggest that all strata within the Olla field have been affected by injection.

As a result, we conclude that all the samples from the Shelton paper have seen injected CO₂ and can be compared.

Line 90: Again, this isn't true. Only the 2800' sand of the Olla Oil Field was injected with CO₂. See Shelton et al., 2014.

As with previous comments focussed on this concern, our findings suggest that all strata within the Olla field have been affected by injection.

Line 94,96,97,102: *G*roup

Capitalised for all.

Line 100: bodie*s*

Changed

Line 104-106: Yes, so you have this information!! This is critical and needs to be addressed. Where did your samples come from???

Producing formation information has been added to Table 1.

Line 113: change a to one

Changed

Line 114: Here is where you falter again. I think it is safe to say (and Shelton et al., 2014 had evidence of this also) that the 2800' sand was not uniformly flooded, not the entire oil field.

As with previous comments focussed on this concern, our findings suggest that all strata within the Olla field have been affected by injection.

In this sentence we do not state that the CO₂ flooding was not uniform, but that the CO₂ is not uniformly distributed within and across the different reservoirs.

Line 121: Or water flooding?

Water flooding is a type of enhanced oil recovery, and is already included within this statement.

Line 136-141: remove capitals after each point number

Changed to lowercase.

Line 161: change 'of the sample tested. This interpretation is based on' to 'of the samples tested, based on'

Change accepted.

Line 170: It does not appear you sampled the Black Lake Field. Where did you get these data?

Reference added

Line 180: *s*amples

Accepted

Line 202: You need a closing parenthesis and I am not sure where it should go here.

Closing parentheses now added. Thank you for spotting this.

Line 206: $^{40}\text{Ar}/^{36}\text{Ar}$ *values*

Added

Line 212: McIntosh et al., 2010 and Shelton et al. work in the Olla Field found geochemical differences within the Wilcox Group, all producing from the Olla Field. Therefore, this is not necessarily an accurate assumption.

We have changed this to read within the Middle Wilcox Group. If the geochemical differences that have been previously observed had affected the $^{40}\text{Ar}/^{36}\text{Ar}$ ratio, we would expect to see variations between the producing zones at Olla which is not observed.

Line 223: How can you make this statement given you do not have noble gas isotope data for the Black Lake Field?

As stated above, we see a significant difference between the noble gas signatures (helium isotopes: $t_{\text{test}(11)} = 28.16$, $p\text{-value} = 1.3 \times 10^{-11}$) at Olla and Nebo Hemphill, with the Olla field displaying a much higher mantle fluid contribution. We also found that samples with the highest CO_2 concentrations have $\text{CO}_2/{}^3\text{He}$ ratios within the mantle range. There is no structure in the region that could explain the enhanced mantle component in the Olla field, but not the Nebo field, as the only significant difference between the Olla and Nebo Hemphill is the presence of injected CO_2 . Noble gas data from close to the Black Lake Oil Field has shown the Sabine Island Uplift (for which the Black Lake Oil Field lies on the edge) is a conduit to mantle fluids in the region (Byrne et al., 2020). We agree that by having noble gas data from the Black Lake Field would further strengthen our argument, however using the noble gas and $\text{CO}_2/{}^3\text{He}$ ratio from this study and previous studies near the Sabine Island Uplift, are fully consistent with the injected fluids having a mantle signature.

Line 225-226: I disagree. What if the injected CO_2 currently isn't impacting the field at all? And because you have no noble gas isotope evidence from the Black Lake Field, you cannot assume the noble gas characteristic of the injected gas.

The same reply as comment above.

Line 242: *G*roup

Changed

Line 249-250: Add: "as also suggested by previous work in the Olla oil field⁵."

Statement reflecting previous work has been added

Line 252: change system to oil field

Changed

Line 289: The Olla oil field isn't a reservoir, it is an oil field.

Changed to reservoirs to reflect multiple producing formations

Line 314-315: Did you measure these data? I do not see this in your tables presented. Cite the McIntosh and Shelton work if you are using their data to make your conclusions.

References have been added for the values taken from Shelton et al 2014. We have also added SOM Table 3 with the formation water chemistry data collected in our study.

Line 335: Did you perform metagenomics? All I have seen is 16S rRNA gene sequencing.

Thank you for highlighting this. Metagenomics has been replaced with 16S rRNA gene sequencing.

Line 350-351: The Shelton et al. work also proposed methanogenesis in the Nebo-Hemphill field.

Apologies for the oversight, reference to the Shelton et al., 2014 paper has now been made within the statement.

Line 352: Just because there was no injected CO₂ indicates that the rates and extent of methanogenesis is different in the Nebo-Hemphill? I do not follow this logic.

We have removed reference to the rate of methanogenesis in Nebo-Hemphill.

Line 384-386: This is a great conclusion and excellent application of this work.

Thank you.

Referee #4 (Remarks to the Author):

I have reviewed with great interest the paper submitted to Nature by Tune et al. The paper is very well written and documented, and importantly it tackles a very important societal issue using a true multi-proxy approach. The field of carbon capture and storage (CCS) is in my opinion a key technology to mitigate anthropogenic greenhouse gases. The main challenges of CCS are finding secure storage locations, and monitoring for CO₂ escape. What this paper does is novel: it asks the question of the impact of the deep biosphere on the fate of CO₂. It does this by comparing two nearby oil fields, one produced with enhanced oil recovery (EOR, the Olla Oil Field) using CO₂ injection, and one (Nebo-Hemphill oil field) where CO₂ injection has never been used. The paper confirms that CO₂ has been trapped in the field produced with CO₂ EOR (i.e. residual trapping) but what is more impactful is that the authors also find that the proportion of CH₄ (methane) is higher in this field. By using stable isotopes, noble gases, organic clumped isotopes and genome sequencing, the authors demonstrate (in my opinion quite convincingly) that the CH₄ is of biogenic origin and is the result of the degradation of CO₂ by bacterial processes.

This has some tremendous implications: CH₄ is an even more potent (albeit short-lived) greenhouse gas in the atmosphere, and its flow properties in the subsurface (and thus the ability of a cap rock to retain it) are very different than that of CO₂. I completely agree with the authors that this has been overlooked, but needs to be taken into consideration for CO₂ injections in future!

I don't have major comments on the paper itself, it is very well written, the data and methods are well explained, and I cannot find flaws in any of the arguments (my expertise is in carbonate clumped isotopes, so perhaps an organic geochemist will have a different view here. But for me it sounds all very reasonable).

Perhaps the only thing I would have liked is for the authors to add one paragraph at the end that spells out the implications of CO₂ being transformed into CH₄ via biogenic pathways during CCS: it will not be clear to all readers of Nature what the challenge of this could be.

We thank the reviewer for their glowing support and agree about the broad impact and importance of this study. We have added a few sentences in our final paragraph stating that CH₄ is less soluble and more mobile than CO₂ and thus has a lower trapping efficiency and thus an increased risk of gas loss. We hope this improves the clarity of the importance of this study.

Reviewer Reports on the First Revision:

Referee #1 (Remarks to the Author):

The manuscript "Rapid microbial methanogenesis during CO₂ storage in hydrocarbon reservoirs" explores to what extent CO₂ injected into an oil reservoir in the 1980's has been converted into methane by microbial methanogenesis. The authors conclude that 13-19% of the CO₂ was converted into methane over 40 years. To make their case, they present methane concentrations, delta13C isotope compositions for methane and CO₂ in sampled gas, clumped isotope composition for methane as well as CO₂/3He ratios.

Conversion of a significant part of CO₂ to methane in CCS trials would be a big deal, both because of the higher warming potential and larger mobility of methane in the subsurface. This study will inform societal debate on trajectories toward "net-zero" in 2050 (to what extent should CCS be implemented?) as well as site selection and monitoring requirements.

The manuscript has come a long way in this revision. Many aspects of the study have become clearer and overall the story has become more convincing in my mind. On the downside, improved clarity has brought up a few new issues, I apologize that I overlooked those in my previous review report.

Major points

(1) I think it would be very helpful to create an image with a schematic of the reservoir and all the relevant processes that the authors discuss: Mass transport processes such as movement of the free gas and groundwater carrying dissolved CO₂ out of the reservoir. Dissolution of CO₂, precipitation. Methanogenes (indicating hydrocarbons as the H₂ donor) and AOR (with sulfate or oxidized manganese or iron). Perhaps, they could even visualize their four arguments/observations that show that methane is being produced: (1) increase of He over CO₂ (2) thermodynamic equilibration of isotopes associated with microbial methane cycling etc. I think such as figure showing the key processes will really help boost the impact of this study. Perhaps the journal could help with that visualization or include it in a News and Views perspective.

(2) When performing 16S amplicon sequencing with separate sets of Archaeal and Bacterial primers, the data does not directly show, as I commented previously, the ratio of archaea to bacteria. However, I realized there is one aspect of the data that actually provides some qualitative insight into this and the authors might investigate their data for this signal. When bacteria are much more abundant than archaea, the archaeal primers will start to pick up bacterial sequences. And vice versa. So could the authors quantify and share whether the two oil fields show any difference in the fraction of bacteria in the archaeal dataset and the fraction of archaea in the bacterial dataset. This could show if there is any enrichment of archaea in the Olla oilfield. Which I would expect, as so much more methane is produced in Olla than in Nebo?

(3) Amplicon sequencing results clearly show prevalence of H₂/CO₂ methanogens. This could be made more clear in the supplementary figure, for example by showing H₂/CO₂ methanogens taxa in a different color or bold faced font.

(4) The figure shows very few ANME, indicating that rates of AOM are likely very low, especially as ANME bacteria have lower rates per cell than methanogens. So the results indicate that the rate of methanogenesis is likely at least 3 orders of magnitude larger than the AOM rate. Even if methanogenesis will proceed in the reverse direction too, as discussed by the authors, the reversed rate is also extremely low compared to the forward rate.

The question is: Is the overall AOR/reversed rate still sufficient to explain the observed equilibration of the isotopes? I think that should at least be discussed.

(5) I did not find the NCBI ID of the amplicon sequencing results. They should at least be submitted to a public repository and a link should be provided. It is surprising how many studies still get away with not sharing their raw sequencing data despite the crackdown on this practice of journals like Nature. Perhaps the authors have done this, I could not find it.

Minor comments

L54-55 "Microbial methanogenesis usually operates by either acetate fermentation or CO₂ consumption". This sentence suggests that methylotrophic methanogenesis is somehow unusual, which is probably not the author's intention. I would rephrase to "Microbial methanogenesis takes acetate, methylated compounds or hydrogen + CO₂ as its starting point;"

L58 "but evolves to an enriched ¹³C content with continued CO₂ consumption" This can be interpreted to mean that the methane becomes more enriched than the source CO₂. Perhaps rephrase to something like "with delta¹³C eventually increasing to the value of the consumed CO₂" or similar.

L97 "provides a conservative estimate of post-injection CO₂ removal in the majority of samples of between 39-89 %" Perhaps more precision could be used here, replace "removal" with "consumption" or "reaction". Removal could also mean efflux out of the reservoir and is thus less precise.

L120/121 "Independent of the source of the injected CO₂, such positive δ¹³C values suggest significant modification from a starting composition that is unlikely to be heavier than 2 ‰_{21,2}" Rephrase for clarity. Especially, what is meant with "modification"? Replace with "reaction", "consumption"? Not sure.

L122 "hydrocarbon biodegradation and subsequent consumption" What do the authors mean here? Oxidation of reservoir hydrocarbons to CO₂? How can something that is already biodegraded still be consumed?

L130-132 "In the absence of other data, the higher δ¹³C of CH₄ from the Olla samples compared

with data from Nebo-Hemphill could be interpreted as evidence that the Olla Field contains a higher proportion of thermogenic gas” It would be good to explicitly state that the CO₂ injected during EOR in the 1980’s did not also contain any (thermogenic) methane.

L135 “a better-fit model can be formulated” This formulation suggests some kind of statistical analysis. Can the authors perform a statistical test with their data that their model/hypothesis fits the data better (for example has higher likelihood) than the alternative explanation (flow of more thermogenic methane into the reservoir).

L135-137 If such a test could be performed, the authors can be more assertive here. Change “Olla is assumed to have had a pre-EOR composition resembling Nebo-Hemphill, but following CO₂-EOR, microbial activity converted significant amounts of injected CO₂ to CH₄” to “Olla had a pre-EOR composition resembling Nebo-Hemphill, but following CO₂-EOR, microbial activity converted significant amounts of injected CO₂ to CH₄.”

L137-138. It also explains where the enormous amount of methane in Olla came from. L73-75: Concentration is slightly lower but because there is so much more gas, still way more methane.

L140-151 This paragraph should describe the authors first argument on why they believe microbial activity converts CO₂ to methane. Instead, the paragraph argues that the isotope data for both reservoirs are at equilibrium at reservoir conditions. I would expect a final sentence that mentions how equilibrium is evidence for microbial activity because this is counterintuitive. I would expect fractionation during methanogenesis to increase the difference in delta¹³C between CO₂ and methane. Perhaps it should also be stated explicitly that both oilfields show evidence for methanogenesis, the difference is more quantitative (rate of methanogenesis) than qualitative (whether or not it occurs).

L156 “Clumped isotope compositions provide a unique and independent constraint” This is an example of misuse of “unique: which is very common in science writing today. Change to “Clumped isotope compositions provide an independent constraint...”

L194-96 “As well as ANME’s being associated with AOM, it has been proposed that some methanogens are also capable of performing methanogenesis in reverse³⁹” This formulation is misleading, because ANMEs also perform methanogenesis in reverse. I wonder if this sideline is not better left out. Better to simply state “Detection of 16S rRNA gene sequences affiliated with known anaerobic methanogens presents strong evidence for the occurrence of AOM in the reservoir.” Next sentence better to state more clearly: “Co-occurrence of both methanogenic and methanotrophic Archaea shows the potential for conversion of CO₂ to methane and vice versa, consistent with isotopic signatures at thermodynamic equilibrium.

L200-202 Change “The integration of independent geochemical and microbiological constraints we provide here overwhelmingly supports our assertion that injected CO₂ is being consumed at Olla by microbial methanogenesis.” to “Combined independent geochemical and microbiological data supports our assertion that oilfield microbes convert injected CO₂ to methane at Olla oilfield.” “Overwhelmingly” is for the reader to decide.

L218-19 “Our combined noble gas and stable isotope approach in the Olla Field allows estimates of in-situ microbial methanogenesis rates...” add “net”, as the gross rate must be higher. After all, there is also AOM that must be compensated.

L224-25 “73-109 mmol CH₄/m³(STP)/yr”. Assuming a rate of 1 fmol/cell/day for both ANME and methanogens (<https://link.springer.com/article/10.1007/s00253-017-8416-0>), 100 mmol CH₄/m³/year requires in the order of 10⁵ relevant cells per ml. I think that is a reasonable number, so at least it passes this basic feasibility test.

L226. Reference 40 does not describe a culture, but an incubation. Change to "hydrogenotrophic methanogens in lab microcosm incubations..." By the way, results in ref 40 clearly indicated that something was inhibiting or limiting methanogenesis in those incubations. A simple comparison as made by the authors is still justified.

Referee #2 (Remarks to the Author):

The authors have responded thoughtfully and in detail to the remarks of all reviewers. In particular, they took my remarks fully into account and in the process corrected some minor mistakes and improved the clarity of the description at some points that I had previously found a bit difficult to understand. Technically, the manuscript has now matured to the point where it is ready for publication. During my careful reading I only detected a few minor mistakes in the supplementary information (missing superscripts for a reference and for ion charges in the first lines of SI.3, a missing space in line 2 of SI.5). This can be resolved in the production stage.

My impression is that the authors did also respond convincingly to more fundamental concerns of some reviewers regarding the importance and novelty of the study. In my view, the study is highly relevant for CCS, which is now also mentioned more clearly in the concluding paragraph. This relevance together with the novel aspects and findings regarding methanogenesis warrants publication of the paper in Nature.

In summary, I recommend publication of the manuscript in present form.

Reviewer: Werner Aeschbach

Referee #3 (Remarks to the Author):

Many thanks to the authors for thoroughly addressing my comments. I can now recommend this article for publication.

However, my concern about CO₂ gas that had been injected migrating to strata OLDER than the 2800' sand (i.e., the CO₂ migrated into deeper strata instead of migrating upwards, which doesn't make sense given injected CO₂ gas would rise) still leaves me puzzled. Can the authors provide the mechanism for which CO₂ gas would migrate to strata deeper than the injected strata? Is there evidence for equal pressures across these strata? One does not typically expect injected CO₂ to leak into underlying reservoirs, so a mechanism for this process needs to be explained in the paper.

Furthermore, the logic that "if Olla Field had previously produced this high of concentrations of CO₂, it would have been used as a source of CO₂" is just a little illogical. The Black Lake Field has a CO₂ mol% of what? How does that compare to the Olla? Are there any fields that produce at 20-10% CO₂ that are used as a source for CO₂ EOR? I do not know of any. So your logic that "CO₂ would not have been injected for EOR into this field if the CO₂ concentrations were naturally this high" is wrong -- CO₂ concentrations this high are commonly observed in methanogenic reservoirs. Therefore, prior to CO₂-EOR, the reservoir could have been exceptionally methanogenic.

These two points should be addressed and, if so, I recommend this manuscript for publication.

Author Rebuttals to First Revision:

Referee #1 (Remarks to the Author):

The manuscript "Rapid microbial methanogenesis during CO₂ storage in hydrocarbon reservoirs" explores to what extent CO₂ injected into an oil reservoir in the 1980's has been converted into methane by microbial methanogenesis. The authors conclude that 13-19% of the CO₂ was converted into methane over 40 years. To make their case, they present methane concentrations, delta¹³C isotope compositions for methane and CO₂ in sampled gas, clumped isotope composition for methane as well as CO₂/³He ratios.

Conversion of a significant part of CO₂ to methane in CCS trials would be a big deal, both because of the higher warming potential and larger mobility of methane in the subsurface. This study will inform societal debate on trajectories toward "net-zero" in 2050 (to what extent should CCS be implemented?) as well as site selection and monitoring requirements.

The manuscript has come a long way in this revision. Many aspects of the study have become clearer and overall the story has become more convincing in my mind. On the downside, improved clarity has brought up a few new issues, I apologize that I overlooked those in my previous review report.

We are glad that the reviewer finds the clarity of the manuscript has been improved and we thank all reviewers in helping us achieve this through both this and previous review rounds.

Major points

(1) I think it would be very helpful to create an image with a schematic of the reservoir and all the relevant processes that the authors discuss: Mass transport processes such as movement of the free gas and groundwater carrying dissolved CO₂ out of the reservoir. Dissolution of CO₂, precipitation. Methanogenes (indicating hydrocarbons as the H₂ donor) and AOR (with sulfate or oxidized manganese or iron). Perhaps, they could even visualize their four arguments/observations that show that methane is being produced: (1) increase of He over CO₂ (2) thermodynamic equilibration of isotopes associated with microbial methane cycling etc. I think such as figure showing the key processes will really help boost the impact of this study. Perhaps the journal could help with that visualization or include it in a News and Views perspective.

We think this is a great idea and have added the figure as Extended Data Figure 4.

(2) When performing 16S amplicon sequencing with separate sets of Archaeal and Bacterial primers, the data does not directly show, as I commented previously, the ratio of archaea to bacteria. However, I realized there is one aspect of the data that actually provides some qualitative insight into this and the authors might investigate their data for this signal. When bacteria are much more abundant than archaea, the archaeal primers will start to pick up bacterial sequences. And vice versa. So could the authors quantify and share whether the two oil fields show any difference in the fraction of bacteria in the archaeal dataset and the fraction of archaea in the bacterial dataset. This could show if there is any enrichment of archaea in the Olla oilfield. Which I would expect, as so much more methane is produced in Olla than in Nebo?

This is an excellent suggestion by the reviewer, however upon closer look at the sequence data, very low signals were amplified outside of the intended target Domain for both primer sets. Bacterial sequences represented less than ~0.0005% of the Archaeal primer amplified sequences, and Archaeal sequences were more rare in the Bacterial primer amplified dataset. The low amounts of sequences returned are such that while they are present at very low amounts, this could also be from contaminating DNA present in low amounts in DNA extraction kits, and we do not want to speculate that bacterial sequences present could provide insight in to the reservoir communities. Future manuscripts could dive deeper into this study with deep sequencing and alternative primer sets, but this was out of scope for the current work.

(3) Amplicon sequencing results clearly show prevalence of H₂/CO₂ methanogens. This could be made more clear in the supplementary figure, for example by showing H₂/CO₂ methanogens taxa in a different color or bold faced font.

We thank the reviewer for this suggestion to improve the figure clarity. We have updated the figure with the relevant taxa in bold.

(4) The figure shows very few ANME, indicating that rates of AOM are likely very low, especially as ANME bacteria have lower rates per cell than methanogens. So the results indicate that the rate of methanogenesis is likely at least 3 orders of magnitude larger than the AOM rate. Even if methanogenesis will proceed in the reverse direction too, as discussed by the authors, the reversed rate is also extremely low compared to the forward rate.

The question is: Is the overall AOR/reversed rate still sufficient to explain the observed equilibration of the isotopes? I think that should at least be discussed.

The assemblages present are likely becoming more and less abundant dependent on substrate availability, which means the present proportion of each assemblage may not be representative of those since injection. In addition, where the substrate concentrations are low, the enzymatic back flux can increase up to 78% of net AOM, resulting in limited net conversion of CH₄ to CO₂, but allow isotopic equilibrium to be approached. We are excited by the idea to understand the respective rates of aom/methanogenesis but the detail of these dynamics will have to be future work that builds on the concepts and observational constraints in the current work.

(5) I did not find the NCBI ID of the amplicon sequencing results. They should at least be submitted to a public repository and a link should be provided. It is surprising how many studies still get away with not sharing their raw sequencing data despite the crackdown on this practice of journals like Nature. Perhaps the authors have done this, I could not find it.

We thank the reviewer for noting this important omission. The data has now been added to NCBI ID and can be found here: <https://www.ncbi.nlm.nih.gov/bioproject/PRJNA744568>, and noted in the paper data statement.

Minor comments

L54-55 “Microbial methanogenesis usually operates by either acetate fermentation or CO₂ consumption”. This sentence suggests that methylotrophic methanogenesis is somehow unusual, which is probably not the author’s intention. I would rephrase to “Microbial methanogenesis takes acetate, methylated compounds or hydrogen + CO₂ as its starting point;”

This sentence has been modified as per reviewers suggestion.

L58 “but evolves to an enriched ¹³C content with continued CO₂ consumption” This can be interpreted to mean that the methane becomes more enriched than the source CO₂. Perhaps rephrase to something like “with delta¹³C eventually increasing to the value of the consumed CO₂” or similar.

We have modified this sentence to “Hydrogenotrophic methanogenesis results in ¹³C enrichment of residual CO₂ and the formation of CH₄ initially ¹³C depleted compared to the source CO₂, which increases to the value of the consumed CO₂” to improve the clarity as suggested.

L97 “provides a conservative estimate of post-injection CO₂ removal in the majority of samples of between 39-89 %” Perhaps more precision could be used here, replace “removal” with “consumption” or “reaction”. Removal could also mean efflux out of the reservoir and is thus less precise.

‘Removal’ has been replaced with ‘trapping’, mirroring the wording at the beginning of the paragraph.

L120/121 “Independent of the source of the injected CO₂, such positive δ¹³C values suggest significant modification from a starting composition that is unlikely to be heavier than 2 ‰” Rephrase for clarity. Especially, what is meant with “modification”? Replace with “reaction”, “consumption”? Not sure.

With this sentence, we wanted to make the point that a change in the δ¹³C-CO₂ would be required, we have yet to mention any processes and feel that the clarity required by the reviewer becomes apparent later in the paper. We note that we have not changed the word ‘modification’.

L122 “hydrocarbon biodegradation and subsequent consumption” What do the authors mean here? Oxidation of reservoir hydrocarbons to CO₂? How can something that is already biodegraded still be consumed?

Here, we mean that the high hydrocarbon have been biodegraded into CO₂ which in turn is consumed by methanogenesis. We have changed the wording in text to “hydrocarbon biodegradation to CO₂ and subsequent consumption” to encapsulate this.

L130-132 “In the absence of other data, the higher δ¹³C of CH₄ from the Olla samples compared with data from Nebo-Hemphill could be interpreted as evidence that the Olla Field contains a higher

proportion of thermogenic gas” It would be good to explicitly state that the CO₂ injected during EOR in the 1980’s did not also contain any (thermogenic) methane.

We have added a sentence explicitly stating this within the paragraph on injection in the supplementary information.

L135 “a better-fit model can be formulated” This formulation suggests some kind of statistical analysis. Can the authors perform a statistical test with their data that their model/hypothesis fits the data better (for example has higher likelihood) than the alternative explanation (flow of more thermogenic methane into the reservoir).

We have reworded this section to avoid the suggestion that it would be useful to apply a statistical comparison between the two scenarios, one of which is completely incompatible with the observations we present. We appreciate the constructive suggestion by the reviewer here. However, the current setting of the study site on land and in a relatively thermally stable setting would suggest that there has been no recent increase in the level of maturation of the source rock. Any addition of thermogenic gas is therefore unlikely and not consistent with the isotopic signature of the Nebo Hemphill fluids (i.e. the $\delta^{13}\text{C}$ of methane would be anticipated to increase to less negative values with increases in thermogenic methane, which is not observed). Our inference that there is an increase in microbial methane is the simplest and more consistent interpretation of the dataset we present here.

L135-137 If such a test could be performed, the authors can be more assertive here. Change “Olla is assumed to have had a pre-EOR composition resembling Nebo-Hemphill, but following CO₂-EOR, microbial activity converted significant amounts of injected CO₂ to CH₄” to “Olla had a pre-EOR composition resembling Nebo-Hemphill, but following CO₂-EOR, microbial activity converted significant amounts of injected CO₂ to CH₄.”

We appreciate the input from the reviewer here. However, we are not able to state that we know the composition of the Olla reservoir fluids pre-injection given the lack of available data. While we believe it is a very robust assumption, we do not feel that such a change to our language would be appropriate. As such, we have not modified this sentence.

L137-138. It also explains where the enormous amount of methane in Olla came from. L73-75: Concentration is slightly lower but because there is so much more gas, still way more methane.

We thank the reviewer for this positive comment.

L140-151 This paragraph should describe the authors first argument on why they believe microbial activity converts CO₂ to methane. Instead, the paragraph argues that the isotope data for both reservoirs are at equilibrium at reservoir conditions. I would expect a final sentence that mentions how equilibrium is evidence for microbial activity because this is counterintuitive. I would expect fractionation during methanogenesis to increase the difference in $\delta^{13}\text{C}$ between CO₂ and methane.

We expect that the approach to equilibrium is the net effect of both microbial methanogenesis and AOM. We have included this within the paragraph “The carbon isotopes within the system are approaching equilibrium under reservoir conditions as a result of microbial cycling of carbon (i.e., methanogenesis and anaerobic oxidation of methane (AOM), evidence for AOM is in the clumped isotopologues below).” It is possible to have net microbial methanogenesis and the isotopes still have an isotopic approach to equilibrium, as AOM is an inefficient process (e.g., Timmer et al., 2017), meaning bond ordering can occur without net conversion to CO₂.

Perhaps it should also be stated explicitly that both oilfields show evidence for methanogenesis, the difference is more quantitative (rate of methanogenesis) than qualitative (whether or not it occurs).

We have included a sentence in the introduction noting methanogenesis in the Nebo-Hemphill region (“Previous studies suggested microbial hydrocarbon degradation and methanogenesis may occur within both the Olla and Nebo-Hemphill fields”). In addition, we have included a section within the supplementary material regarding methanogenesis at Nebo-Hemphill that includes the statement about the quantitative difference between fields (“however, unlike at Olla, this has not been enhanced by any injection to the field.”).

L156 “Clumped isotope compositions provide a unique and independent constraint” This is an example of misuse of “unique: which is very common in science writing today. Change to “Clumped isotope compositions provide an independent constraint...”

Changed following reviewers suggestion.

L194-96 “As well as ANME’s being associated with AOM, it has been proposed that some methanogens are also capable of performing methanogenesis in reverse³⁹” This formulation is misleading, because ANMEs also perform methanogenesis in reverse. I wonder if this sideline is not better left out. Better to simply state “Detection of 16S rRNA gene sequences affiliated with known anaerobic methanogens presents strong evidence for the occurrence of AOM in the reservoir.” Next sentence better to state more clearly: “Co-occurrence of both methanogenic and methanotrophic Archaea shows the potential for conversion of CO₂ to methane and vice versa, consistent with isotopic signatures at thermodynamic equilibrium.

Changed following reviewers suggestion.

L200-202 Change “The integration of independent geochemical and microbiological constraints we provide here overwhelmingly supports our assertion that injected CO₂ is being consumed at Olla by microbial methanogenesis.” to “Combined independent geochemical and microbiological data supports our assertion that oilfield microbes convert injected CO₂ to methane at Olla oilfield.” “Overwhelmingly” is for the reader to decide.

Changed following reviewers suggestion.

L218-19 “Our combined noble gas and stable isotope approach in the Olla Field allows estimates of

in-situ microbial methanogenesis rates...” add “net”, as the gross rate must be higher. After all, there is also AOM that must be compensated.

Changed following reviewers suggestion.

L224-25 “73-109 mmol CH₄/m³(STP)/yr”. Assuming a rate of 1 fmol/cell/day for both ANME and methanogens (<https://link.springer.com/article/10.1007/s00253-017-8416-0>), 100 mmol CH₄/m³/year requires in the order of 10⁵ relevant cells per ml. I think that is a reasonable number, so at least it passes this basic feasibility test.

We thank the reviewer for doing this feasibility test and positive comment.

L226. Reference 40 does not describe a culture, but an incubation. Change to “hydrogenotrophic methanogens in lab microcosm incubations...” By the way, results in ref 40 clearly indicated that something was inhibiting or limiting methanogenesis in those incubations. A simple comparison as made by the authors is still justified.

Changed following reviewers suggestion.

Referee #2 (Remarks to the Author):

The authors have responded thoughtfully and in detail to the remarks of all reviewers. In particular, they took my remarks fully into account and in the process corrected some minor mistakes and improved the clarity of the description at some points that I had previously found a bit difficult to understand. Technically, the manuscript has now matured to the point where it is ready for publication. During my careful reading I only detected a few minor mistakes in the supplementary information (missing superscripts for a reference and for ion charges in the first lines of SI.3, a missing space in line 2 of SI.5). This can be resolved in the production stage.

My impression is that the authors did also respond convincingly to more fundamental concerns of some reviewers regarding the importance and novelty of the study. In my view, the study is highly relevant for CCS, which is now also mentioned more clearly in the concluding paragraph. This relevance together with the novel aspects and findings regarding methanogenesis warrants publication of the paper in Nature.

In summary, I recommend publication of the manuscript in present form.

Reviewer: Werner Aeschbach

We thank the reviewer for his positive remarks. We have looked through the supplementary information and have corrected the missing superscripts and missing space.

Referee #3 (Remarks to the Author):

Many thanks to the authors for thoroughly addressing my comments. I can now recommend this

article for publication.

However, my concern about CO₂ gas that had been injected migrating to strata OLDER than the 2800' sand (i.e., the CO₂ migrated into deeper strata instead of migrating upwards, which doesn't make sense given injected CO₂ gas would rise) still leaves me puzzled. Can the authors provide the mechanism for which CO₂ gas would migrate to strata deeper than the injected strata? Is there evidence for equal pressures across these strata? One does not typically expect injected CO₂ to leak into underlying reservoirs, so a mechanism for this process needs to be explained in the paper.

We understand where the reviewer is coming from. However, there are several mechanisms whereby fluids in one reservoir can migrate in to overlying and underlying reservoirs. One common mechanism is through fault juxtaposition of an older stratigraphic unit against a slightly younger stratigraphic unit (i.e. one that is often deeper in terms of a vertical component, but laterally can be juxtaposed against it). Such features are common and difficult to identify at a local scale, but represent perhaps the most efficient way for fluids to mix between older and younger lithologies (rather than simply deeper or shallower units). Alternatively, there may be lithologic variability that provides a more efficient pathway for fluids to migrate from a younger reservoir unit in to an older interval. It is also common in subsurface systems for different lithologies to contain different fluid pressures, for example if fluid is extracted from one lithology proximal to another lithology, the pressure in that lithology will be reduced and fluid will migrate from the surrounding lithologies into the reduced pressure lithologies independent of whether they are from above or below this reduced pressure lithology. This is perhaps a slower process than via fault juxtaposition, but also capable of resulting in migration of fluids in to stratigraphically older units. We appreciate that this can be at first glance counter intuitive, but this is a very common occurrence.

Furthermore, the logic that "if Olla Field had previously produced this high of concentrations of CO₂, it would have been used as a source of CO₂" is just a little illogical. The Black Lake Field has a CO₂ mol% of what? How does that compare to the Olla? Are there any fields that produce at 20-10% CO₂ that are used as a source for CO₂ EOR? I do not know of any. So your logic that "CO₂ would not have been injected for EOR into this field if the CO₂ concentrations were naturally this high" is wrong -- CO₂ concentrations this high are commonly observed in methanogenic reservoirs. Therefore, prior to CO₂-EOR, the reservoir could have been exceptionally methanogenic.

We have removed the sentence from the methods section. We think the clarity in our original interpretation remains, namely that concentrations of CO₂ greater than 10% represent breakthrough of injected CO₂.

These two points should be addressed and, if so, I recommend this manuscript for publication.

Reviewer Reports on the Second Revision:

Referee #1 (Remarks to the Author):

All good. I just have one remaining point to complain about. I indicated that the amplicon data showed that the rate of AOM (reversed methanogenesis) is likely 1000x smaller than the forward rate of methanogenesis. Equilibration of isotopes, a key finding of the paper, thus appears to be rate limited by AOM. The authors have addressed this point in their rebuttal, essentially saying that they do not want to interpret their own amplicon sequencing results - but have not made any

changes to the text.

Amplicon sequencing has biases and of course sampling is never going to be extensive in a subsurface habitat, but I believe that an abundance ratio of two related groups of microbes represents a pretty solid result, it doesn't get any better than that.

Given the importance of the isotope equilibration to the argument, I would strongly suggest that the authors make a statement about this in the relevant section of the manuscript because if they don't make a disclaimer, others will consider it an oversight.

Simply stating that this "equilibration appears to be rate limited by AOM, as abundance of methanogens was 100x higher than ANME and AOM likely proceeds at a lower rate per cell" would be sufficient.

Best wishes,
Marc Strous

Referee #3 (Remarks to the Author):

Thank you again for addressing my comments about the Black Lake Field and CO₂ leaking to underlying strata.

I can now recommend this manuscript for publication.

Author Rebuttals to Second Revision:

Referee #1 (Remarks to the Author):

All good. I just have one remaining point to complain about. I indicated that the amplicon data showed that the rate of AOM (reversed methanogenesis) is likely 1000x smaller than the forward rate of methanogenesis. Equilibration of isotopes, a key finding of the paper, thus appears to be rate limited by AOM. The authors have addressed this point in their rebuttal, essentially saying that they do not want to interpret their own amplicon sequencing results - but have not made any changes to the text.

Amplicon sequencing has biases and of course sampling is never going to be extensive in a subsurface habitat, but I believe that an abundance ratio of two related groups of microbes represents a pretty solid result, it doesn't get any better than that.

Given the importance of the isotope equilibration to the argument, I would strongly suggest that the authors make a statement about this in the relevant section of the manuscript because if they don't make a disclaimer, others will consider it an oversight.

Simply stating that this "equilibration appears to be rate limited by AOM, as abundance of methanogens was 100x higher than ANME and AOM likely proceeds at a lower rate per cell" would be sufficient.

We now have added the following sentence at the end of our paragraph on the microbial communities.

“However, equilibration is likely rate limited by AOM, as abundance of methanogens was 100x higher than ANME, and AOM likely proceeds at a lower rate per cell”

Best wishes,
Marc Strous

We thank Prof Strous for his comprehensive and supportive review, which, has greatly improved the manuscript.

Referee #3 (Remarks to the Author):

Thank you again for addressing my comments about the Black Lake Field and CO₂ leaking to underlying strata.

I can now recommend this manuscript for publication.

We thank reviewer #3 for their comprehensive and supportive review, which, has greatly improved the manuscript.